# The molecular organization of differentially curved caveolae indicates bendable structural units at the plasma membrane

Claudia Matthaeus[1], Kem A. Sochacki[1], Andrea M. Dickey [1], Dmytro Puchkov [2], Volker Haucke [2,3], Martin Lehmann [2] & Justin W. Taraska [1]✉

Caveolae are small coated plasma membrane invaginations with diverse functions. Caveolae undergo curvature changes. Yet, it is unclear which proteins regulate this process. To address this gap, we develop a correlative stimulated emission depletion (STED) fluorescence and platinum replica electron microscopy imaging (CLEM) method to image proteins at single caveolae. Caveolins and cavins are found at all caveolae, independent of curvature. EHD2 is detected at both low and highly curved caveolae. Pacsin2 associates with low curved caveolae and EHBP1 with mostly highly curved caveolae. Dynamin is absent from caveolae. Cells lacking dynamin show no substantial changes to caveolae, suggesting that dynamin is not directly involved in caveolae curvature. We propose a model where caveolins, cavins, and EHD2 assemble as a cohesive structural unit regulated by intermittent associations with pacsin2 and EHBP1. These coats can flatten and curve to enable lipid traffic, signaling, and changes to the surface area of the cell.

Caveolae are 60–100 nm diameter coated plasma membrane domains that invaginate into the cytosol. They are prominent and common features of the plasma membrane in many cells including adipocytes, fibroblast, and muscle cells[1]. They concentrate specific lipids including cholesterol, sphingolipids, and PI(4,5)P$_2$, supporting the clustering of distinct proteins and signaling molecules[2,3]. Mice lacking caveolae have defects in lipid uptake, blood vessel function, and membrane tension regulation[4]. Dysregulation or mutation of caveolae proteins are known drivers of diseases including muscle disorders[5,6] and cancer[7].

Much is known about the molecular components of caveolae. The caveolin proteins (caveolin1–3 in mammals) and cavins (cavin1–4) are central for organelle formation. Deletion of either caveolin1 or cavin1 results in loss of caveolae in vivo[4]. Structural data indicate that cavins form hetero-trimers (mainly cavin1/2 or cavin1/3) and assemble as a layered protein coat with caveolins[8–11]. Membrane remodeling proteins such as Eps15 homology (EH) domain-containing protein 2 (EHD2)[12,13], pacsin/syndapin2[14–16] and EH domain-binding protein 1 (EHBP1[17]) also associate with caveolae. In particular, the ATPase EHD2 plays an

important role in stabilizing caveolae at the plasma membrane and regulating endocytosis[12,13]. A number of studies have used knockdown or mutations to identify roles for these factors in caveolae function[18–23]. For example, loss of EHD2 in vivo did not change caveolae assembly and number. However, an increased mobility of caveolae was detected in mice lacking EHD2 that was associated with an increase in lipid accumulation[24]. EHD2 has been localized to the caveolae neck with immunogold electron microscopy (EM) of thin sections[25]. Based on structural data, EHD2 has been proposed to specifically form a ring-like oligomer encircling the caveolar neck[26]. The BAR domain-containing protein pacsin2 was also found at the caveolae neck[16]. Deletion of the muscle-specific variant pacsin3 led to a loss of the characteristic caveolae bulb shape despite the fact that caveolin1 and cavin1 were still present at the plasma membrane[27]. Knockdown of pacsin2 leads to shallow caveolae, and impairs caveolae mobility and endocytosis[14,16,28]. Recently, EHBP1 was found to stabilize caveolae at the plasma membrane. Loss of EHBP1, similar to loss of EHD2, however, did not modulate caveolae shapes but rather increased endocytosis[17]. In addition,

[1]Biochemistry and Biophysics Center, National Heart, Lung, and Blood Institute, National Institutes of Health, Bethesda, MD, USA. [2]Leibniz-Forschungsinstitut für Molekulare Pharmakologie (FMP), Berlin, Germany. [3]Faculty of Biology, Chemistry and Pharmacy, Freie Universität Berlin, Berlin, Germany. ✉e-mail: justin.taraska@nih.gov

dynamin has been commonly implicated in caveolae endocytosis and proposed to have a similar role to its well-established functions in membrane scission in clathrin-mediated endocytosis[29–31]. Other notable proteins have been linked to caveolae including receptor tyrosine kinase-like orphan receptor 1 (ROR1) in embryonic tissues[32] or the c-Abl tyrosine kinase FBP17 in rosette-like caveolae clusters[33,34]. Yet, while much data is known, it is unclear how these components assemble and curve at the plasma membrane[35,36]. Understanding this architecture is necessary for understanding how caveolae function in cells, what their roles are, and how they are regulated across different pathways and tissues.

The structures of purified caveolin1 complexes suggest that the assembly of 11 caveolin1 molecules into ~14 nm disc-shaped oligomer is needed to induce caveolae formation[37–39]. Cavins are then recruited to these sites leading to substantial membrane bending. This process is proposed to be reversible during increased membrane tension (e.g., osmotic shock) or cellular stress (e.g., UV light) where cavins are thought to be released, leading to a flattening of the invagination[10,40,41]. However, it is currently unknown how or if flat and highly curved caveolae differ in their morphologies and protein components. In addition, it is unclear if cavins are released during flattening, if caveolae disassemble upon increased membrane tension, or if caveolae exhibit a more flexible coat that, similar to clathrin-coated pits, can change its shape from flat to curved. Furthermore, it is not known when, and how the caveolae proteins EHD2, pacsin2 and EHBP1 are recruited to caveolae membrane domains and how these proteins regulate the caveolin/cavin coat complex and its curvature. As an example, EHD2 was shown to translocate into the nucleus after caveolae flatten[42] indicating that EHD2 may not associate with flat caveolae. A detailed understanding of the coat and associated proteins, however, in relation to caveolae shape and curvature is missing. These questions have been difficult to answer with light microscopy due to the small size of caveolae relative to the diffraction limit. Also, past EM measurements were not optimal for two reasons. First, imaging all caveolae in a membrane to provide a population-level structural view in thin section EM is challenging[43], and second, localizing and quantifying specific protein components within those EM images is difficult with established labeling and analysis methods[25] in the context of different caveolae curvature types.

To overcome these limitations, we studied the relationship between caveolae morphology and the key proteins proposed to regulate caveolae structure and behavior with nanoscale correlative light and EM across entire plasma membranes. First, to understand the shape of caveolae across many single cells, we use platinum replica electron microscopy (PREM) to classify, analyze, and quantitate caveolae membrane domains at the plasma membrane into low, medium, and highly curved caveolae. Next, to understand how proteins associate with these shapes, we developed a super-resolution STED and platinum replica correlation method (STED-CLEM) to localize major caveolae coat and regulatory proteins in and around single caveola across entire plasma membranes of cultured mammalian cells. Surprisingly, different from previous models, we find that along with caveolins, EHD2 and cavins were present on both low and highly curved caveolae, while EHBP1 was mainly found at a subset of highly curved caveolae. Pacsin2 was primarily detected at low curved caveolae. Dynamin was absent from caveolae. Loss of these proteins differentially affected caveolae shape and abundance. Taken together, we present direct nanoscale insights into the control of caveolae curvature across the plasma membrane and propose a new molecular model for the regulation of caveolae curvature in mammalian cells.

## Results

### Structural investigation and classification of caveolae curvature
Caveolae coats are proposed to change their curvature depending on membrane tension[44] and maturation[36]. This process is not understood.

In particular, how flat caveolae curve, or how curved caveolae flatten, and which proteins regulate these transitions are unclear. To gain a global view of caveolae density, shapes, and sizes, we analyzed caveolae at the plasma membrane across several common cultured cell types with PREM. To visualize caveolae at high resolution, cells were grown on coverslips, unroofed with a light sheering force to expose their inner plasma membranes, fixed, and platinum replicas of the cytosolic face of these membranes were generated and imaged by transmission electron microscopy (TEM, Fig. 1a)[45,46]. In these images, caveolae can be identified by their size, round shape, and distinctive watermelon-like striped coat (orange arrows in Fig. 1a). Adipocytes contain substantially more caveolae at their plasma membrane compared to other cell types (Fig. 1b, caveolae number/$\mu m^2$: MEF 2.6 ± 0.4, adipocytes 14.8 ± 1.4, myoblasts 1.7 ± 0.5, astrocytes 3 ± 0.8, HUVEC 2.4 ± 0.3, HeLa 1.9 ± 0.3). Caveolae had diameters between 40 and 160 nm. In mouse embryonic fibroblasts (MEFs) and endothelial cells (Human umbilical vein endothelial cells, HUVEC) caveolae diameters were slightly larger compared to other cell types (Fig. 1c, MEF 100.5 ± 2.1 nm, adipocytes 84.9 ± 1.3 nm, myocytes 84.5 ± 1.6 nm, astrocytes 83.2 ± 1.6 nm, HUVEC 98.5 ± 1.6 nm, 80.2 ± 1.2 nm). Notably, rosette-like caveolae cluster were found in various shapes and dimensions in PREM images (Suppl. Fig. 1). The caveolae coat contained an average of 4–5 visible coat "stripes" with a length of 47.5 ± 1.2 nm (Fig. 1c, Suppl. Fig. 2a) and a convoluted spiral-like topology with an average mean distance of 16.2 ± 0.5 nm between each stripe (Suppl. Fig. 2b, c).

To verify that these structures were caveolae, the caveolae proteins cavin1 (Fig. 1d) and caveolin1 (Fig. 1e), were labeled with antibody or NTA-linked nanogold particles and imaged with platinum replica EM. Figure 1d shows plasma membranes of MEFs expressing His-tagged cavin1 treated with nickel-NTA labeled gold particles. Caveolin1 was detected by antibody labeling. Both could be observed at caveolae. Thus, cavin1 and caveolin1 mark morphologically-identifiable caveolae. They also associate with small and disorganized coat domains with low curvatures (Fig. 1d left panel).

Next, we quantitively analyzed caveolae curvature in different cell types. Platinum replicas of unroofed MEFs, HUVEC, and HeLa cell inner plasma membranes were generated, imaged, and segmented for all visible caveolae. Figure 2a shows example platinum replica EM images of caveolae with a range of morphologies from low to highly curved invaginations. Figure 2b shows example caveolae curvature types across the three different cell types. The size, packing, and arrangement of the coat varied. Visible stripes could be seen on single caveolae. Low curved caveolae exhibit a close packing of the coat and low curvatures (Fig. 2a). Medium invaginated caveolae showed a similar coat texture, however, the organelle coat was more curved, forming a bulb shape with a noticeable edge density (white signal) relative to the center of the organelle. This edge signal arises from metal accumulated along the side of the organelle when the sample is coated with platinum at an angle. The thicker material blocks passing electrons and appears as a white ring in inverted images. Highly curved caveolae have an even stronger edge signal with no clear membrane transitioning from the caveolar body to the surrounding membrane. From these criteria, we classified caveolae into three general morphological classes: low, medium, and highly curved. Of note, in highly invaginated caveolae, the caveolar neck is located under the coat and is therefore hidden when viewed from above[47]. Thus, we cannot ascertain if highly curved caveolae membranes are connected or separated from the plasma membrane. In support of these 2D PREM images, electron tomograms of platinum replicas were acquired (Fig. 2c, Video S1–4). Low curved caveolae showed a lower height and lower curvature compared to medium and highly curved caveolae (Fig. 2c, d, Video S1–4).

To confirm that caveolae could be manually grouped into these classes, we measured the average intensity projections of

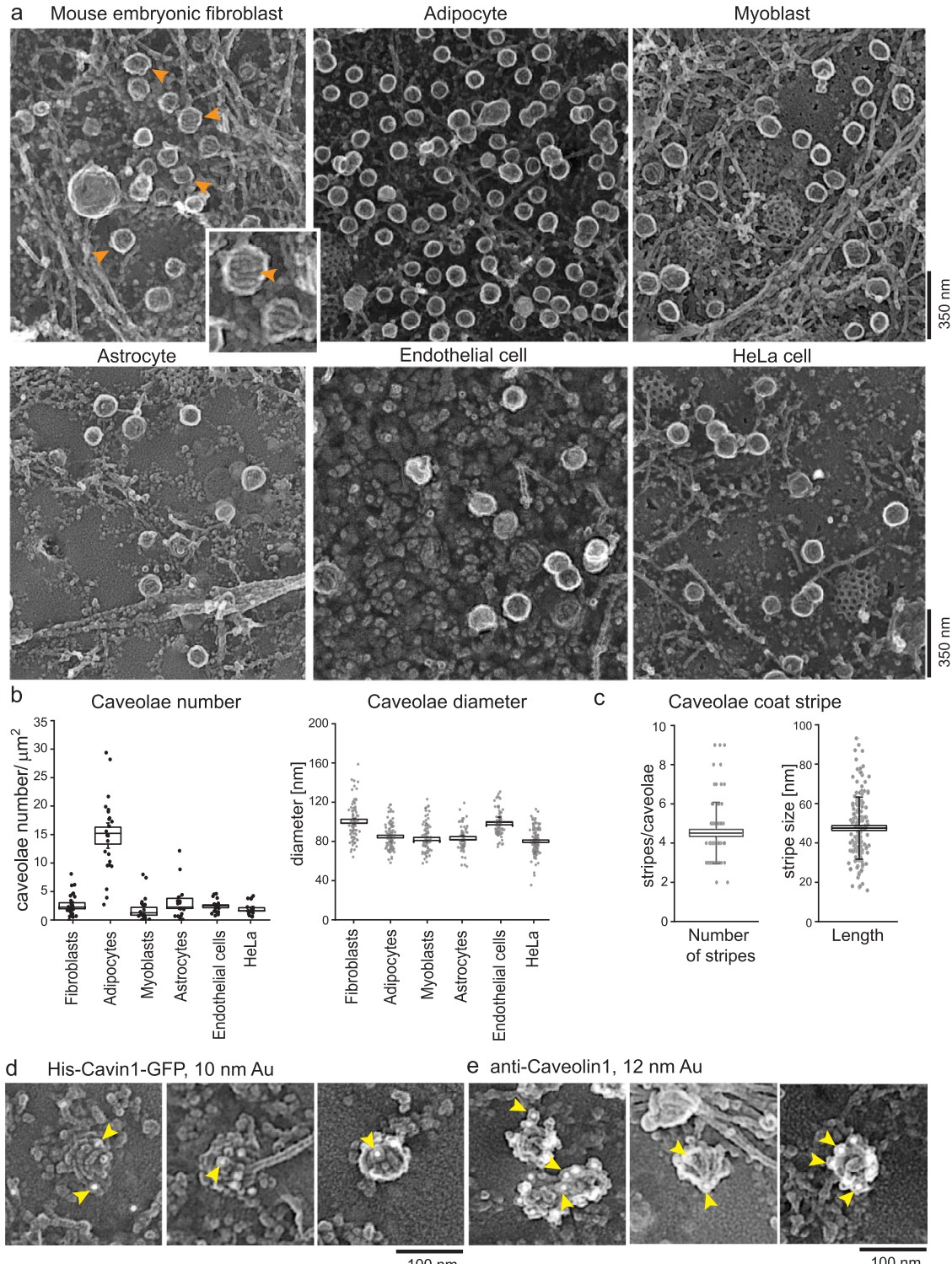

**Fig. 1 | Overview of caveolae at the plasma membrane in various cell types.**
**a** Representative platinum replica transmission electron microscopy (PREM) images of plasma membrane sheets of different cell types. The images are presented in an inverted scale. Orange arrows indicate individual caveolae. **b** Caveolae number and diameter were measured in all cell types (n(fibroblasts) = 26, n(adipocytes) = 22, n(myoblasts) = 22, n(astrocytes) = 17, n(endothelial cells) = 21, n(HeLa) = 19; diameter: n(fibroblasts, MEFs) = 82, n(adipocytes) = 95, n(myoblasts) = 83, n(astrocytes) = 64, n(endothelial cells, HUVEC) = 70, n(HeLa) = 118, 2 independent

experiments). Box plots represent median values, bounds of box represent ± SE, whiskers show SD, each replicate is depicted. **c** Number of single coat stripes per caveolae (MEF, *n* = 79), and stripe length in nm (*n* = 162) of single coat stripes. Box plots represent mean values ± SE, whiskers show SD, each replicate is depicted. **d,e** Identification of cavin1 (**d**) or caveolin1 (**e**) in PREM of MEFs. His-Cavin1-EGFP was expressed in MEFs and 10 nm Ni-NTA nanogold was used to labeled His tags. Caveolin1 was investigated by immunolabeling and 12 nm gold secondary antibody. Yellow arrows indicate gold particles (2 independent experiments).

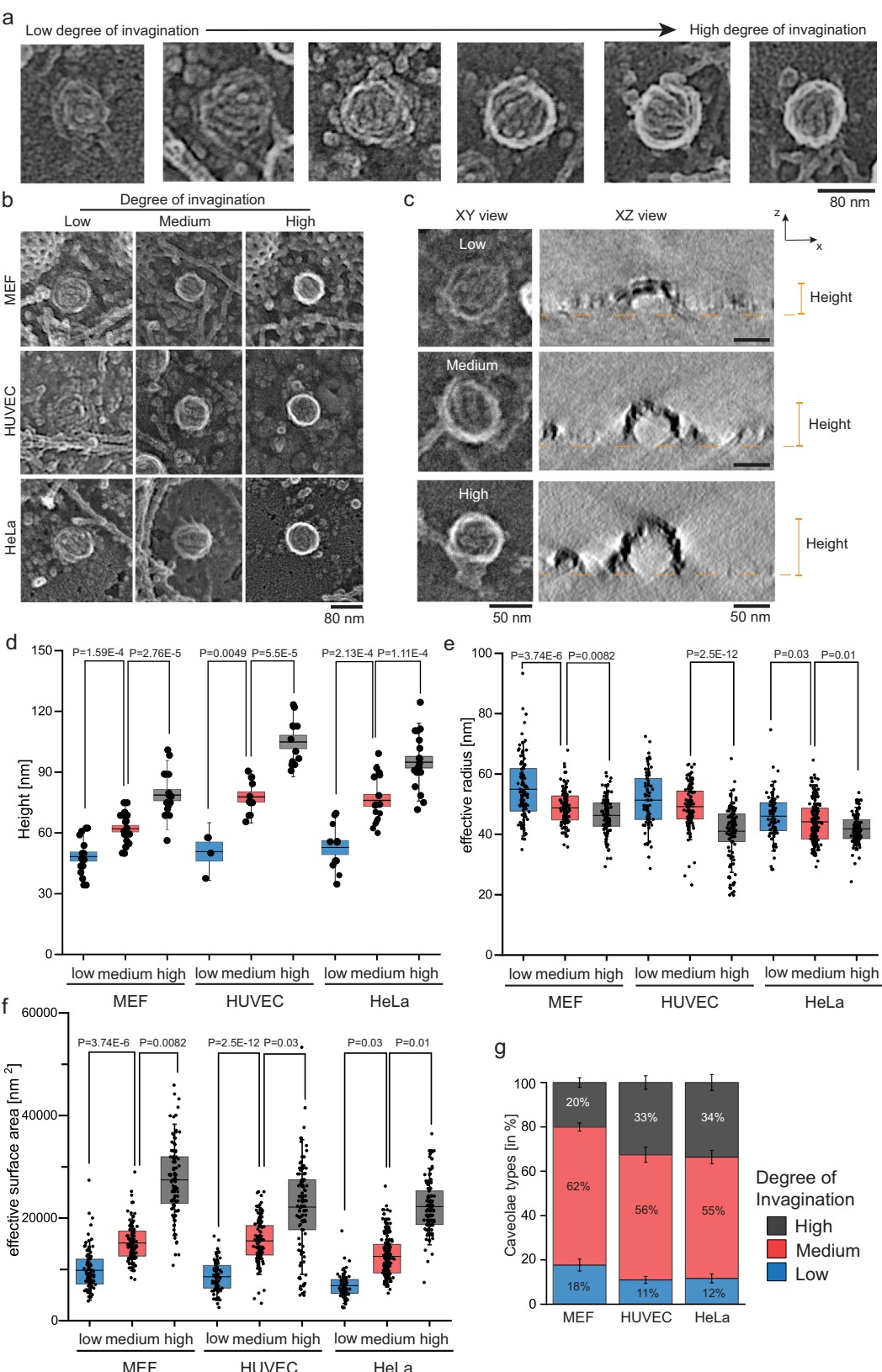

low, medium, or highly curved caveolae groups (Suppl. Fig. 3a). When line scans of the gray value profile were normalized and plotted from the center of the individual organelles toward their edges, the averages of different caveolae types could be distinguished (Suppl. Fig. 3b). Low curved caveolae exhibited minimal intensity difference on average between the center, edge,

and outside of the organelle image (blue graph in Suppl. Fig. 3b). Yet, a clear edge signal was detectable on average in medium curved caveolae that was reflected in an intensity maximum in Suppl. Fig. 3b (red graph). Highly curved caveolae showed a steeper slope and larger intensity difference that was shifted toward the center of the organelle (black graph in Suppl. Fig. 3b).

**Fig. 2 | Highly curved caveolae show distinct round membrane edges and a smaller size. a** Representative PREM images of caveolae types in which the membrane leaflet can be identified as a dark background and protein by increased electron intensity (gray or white signal, in MEFs). Strong membrane curvature or bending is indicated by a white edge signal. **b** Representative PREM images of low, medium and high degree of caveolar invagination in MEF, HUVEC, and HeLa cells. Scale bar is 80 nm. **c** Representative TEM tomogram xz-images of low, medium and highly curved caveolae obtained from MEF plasma membrane sheets. See also Suppl. Video S1–4. Yellow dashed line marks plasma membrane. **d** Measured height of low, medium and highly curved caveolae in MEF, HUVEC and HeLa tomograms (caveolae number: MEF: n(low) = 16, n(medium) = 21, n(high) = 17; HUVEC: n(low) = 4, n(medium) = 11, n(high) = 12; HeLa: n(low) = 11, n(medium) = 16, n(high) = 21; 3 independent experiments) (**e**) Effective radius of caveolae was calculated with the assumption of round caveolae membrane domains (caveolae number: MEF: n(flat) = 100, n(bulb) = 113, n(sphere) = 96; HUVEC: n(flat) = 67, n(bulb) = 108, n(sphere) = 113; HeLa: n(flat) = 76, n(bulb) = 177, n(sphere) = 133; 3 independent experiments. **f** Effective surface area of flat (based on circle: $A = \pi r^2$), bulb (based on hemisphere: $A = 2\pi r^2$), and spherical caveolae (based on sphere: $A = 4\pi r^2$; caveolae number: MEF: n(flat) = 100, n(bulb) = 113, n(sphere) = 96; HUVEC: n(flat) = 67, n(bulb) = 108, n(sphere) = 113; HeLa: n(flat) = 76, n(bulb) = 177, n(sphere) = 133; 3 independent experiments. **g** Distribution of caveolae types in MEF, HUVEC and HeLa (n(MEF) = 12 cell regions, n(HUVEC) = 13 cell regions, n(HeLa) = 12 cell regions). Bar plot shows mean ± SE. Box plots represent mean values with bounds from 25 to 75 percentage, whiskers illustrate SD, each replicate is depicted. Normal distributed groups were analyzed by two-sided *t*-test, not normally distributed values with two-sided Mann–Whitney test.

The differences in plot profiles were further reflected in the measured gray value difference between maximal and minimal value of the plot profile (Suppl. Fig. 3b, c). The same trend was found in HUVEC and HeLa cells (Suppl. Fig. 3d) further illustrating that strongly curved caveolae can be reliably distinguished from low curved caveolae (Suppl. Fig. 3e). This was further supported by a linear correlation of the plot profile and the measured heights of segmented caveolae determined from 3D electron tomograms of replicas (Suppl. Fig. 3f).

Low curved caveolae were wider than medium and highly curved caveolae, indicating that coat bending decreases the diameter of the caveolae coat (Fig. 2e). Estimated surface areas could be calculated based on the radii[48] where low curved caveolae were assumed as circles, medium as hemispheres, and highly curved as spheres. Given these changes, the calculated surface area of the organelle increased during invagination (Fig. 2f). This is in line with previous observations[34,40] suggesting that caveolae capture additional membrane and swell into the cell, decreasing the cell's exposed surface area when the coat assembles and bends. Importantly, Fig. 2g shows that in all three cell types, caveolae were detected (Fig. 2g). MEFs contained slightly more low and medium curved caveolae compared to HUVEC and HeLa cells (Fig. 2g).

## STED microscopy of single caveola in plasma membrane sheets

To localize caveolae-related proteins on single caveola at the nanoscale, we developed a two-color super-resolution fluorescence microscopy (stimulated emission depletion, STED) method. First, MEF plasma membrane sheets were immuno-stained against caveolin1 (a marker for caveolae, Fig. 3a). As illustrated in Fig. 3b, sufficient lateral resolution could be achieved with the STED dye Atto647N (see also Suppl. Fig. 4a, b) to visualize single caveolin1 spots. Next, two-color STED was used to localize cavin1 in relation to caveolin1 (Fig. 3c, d). To image cavin, cavin1-EGFP was expressed in MEFs and labelled with Atto647-GFP-nanobodies. Caveolin1 was detected by immunolabelling with anti-caveolin antibodies and Alexa594-secondary antibodies (Fig. 3c, d). Compared to the caveolae diameter measured by PREM (101 ± 2 nm, Fig. 1b), the size of caveolin1-positive STED spots (full width at half maximum) labeled with Atto647N showed no substantial size differences across the averages with either GFP-nanobody (114 ± 6 nm) or antibody immunolabeling (108 ± 7 nm, Suppl. Fig. 4c, d). Alexa594 labeling suggested a slightly larger size of 138 ± 7 nm. Further, endogenous levels of cavin1 and caveolin1 were detected by antibody labeling and two-color STED (Suppl. Fig. 4e).

Next, we investigated the localization of the coat proteins caveolin2 and cavins in relation to caveolin1 with two-color STED (Fig. 3e, f). EGFP-tagged caveolin2, cavin1, 2, or 3 were expressed and labelled with a GFP nanobody-Atto647N probe. Caveolin1 was immunolabelled and detected with Alexa594. As expected, cavin coat proteins strongly colocalized with caveolin1 (Fig. 3e). In contrast, the caveolar regulatory proteins EHD2 and pacsin2, as well as EHBP1, exhibited a more punctate localization at caveolin1 spots

(Fig. 3e). This produced a weaker average fluorescence signal relative to the background when many spots were aligned and averaged (Fig. 3f). Cavin1 was detected at more than 90% of all caveolin1-spots, EHD2 at 67% of all caveolin1 spots. Pacsin2 and EHBP1 were observed at 40% or 20% of all caveolin1 spots, respectively (Fig. 3g). These data suggest that both EHBP1 and pacsin2 localize to a subset of caveolae at any given time. Analyzing the average fluorescence profiles of caveolae-associated proteins (Fig. 3h), all caveolae coat proteins shared a similar distribution at STED resolutions compared to caveolin1 (Fig. 3i). EHD2 and pacsin2 exhibited a slightly more extended shape. In summary, STED microscopy can visualize multiple proteins at single caveolae at the plasma membrane. We hypothesized that pacsin2 and EHBP1 may associate with a specific caveolar shape.

## STED Pt replica CLEM reveals cavin localization to low and highly curved caveolae types

To test the hypothesis that specific caveolae related proteins associate to distinct caveolae shapes, we directly correlated STED images to Pt replica EM images of the same samples with a correlative light and electron microscopy method (CLEM[45,46]). Here, caveolae protein localizations could be directly compared to the curvature of the organelle and the surrounding cell membrane (STED-CLEM, Suppl. Fig. 5 overview of correlated MEF). First, MEFs expressing caveolin1-EGFP or caveolin2-EGFP were labeled with a GFP nanobody-Atto647N and imaged (Fig. 4a, b). This approach made it possible to detect caveolin accumulation at all caveolae sub-types (Fig. 4a, b). As the caveolae coat is formed, the caveolin1 STED signals mark morphologically and EM-identifiable caveolae. Notably, the CLEM approach further revealed caveolin1 accumulation at the plasma membrane in non-caveolar sites indicating membrane areas where caveolae may be forming or disassembling (Suppl. Fig. 6a). Caveolin2 showed a similar behavior (Fig. 4b, Suppl. Fig. 6a). Quantitative analysis of the fluorescence associated with the EM segmented caveolae regions indicated that caveolin1 and 2 profiles for low or highly curved caveolae had similar distributions (Fig. 4c, edge of caveolae indicated in green dashed line).

We next imaged cavin1-3 (Fig. 4d–f). We detected the three cavin isoforms in all three caveolae types, similar to data from immunogold labeling (Fig. 1d). Analysis across the entire population of labeled caveolae (n = 488) indicated that cavin1-3 localized with similar distributions to low, medium, and highly curved caveolae (Fig. 4g, h). In addition, cavin1-3 was also observed in non-caveolar sites where the membrane domains were small and disorganized and may be new caveolae formation sites (Suppl. Fig. 6b).

To further confirm this, we imaged endogenous cavin1-3 in Pt replicas of endothelial cells after both mild (1:5 dilution with deionized water) and strong osmotic shock (1:9 dilution with deionized water, accordingly to[40]) with antibody labeling by STED-CLEM (Fig. 5 and Suppl. Fig. 7). Osmotic shock in HUVEC resulted in an increase in lower curved caveolae independent of the intensity of

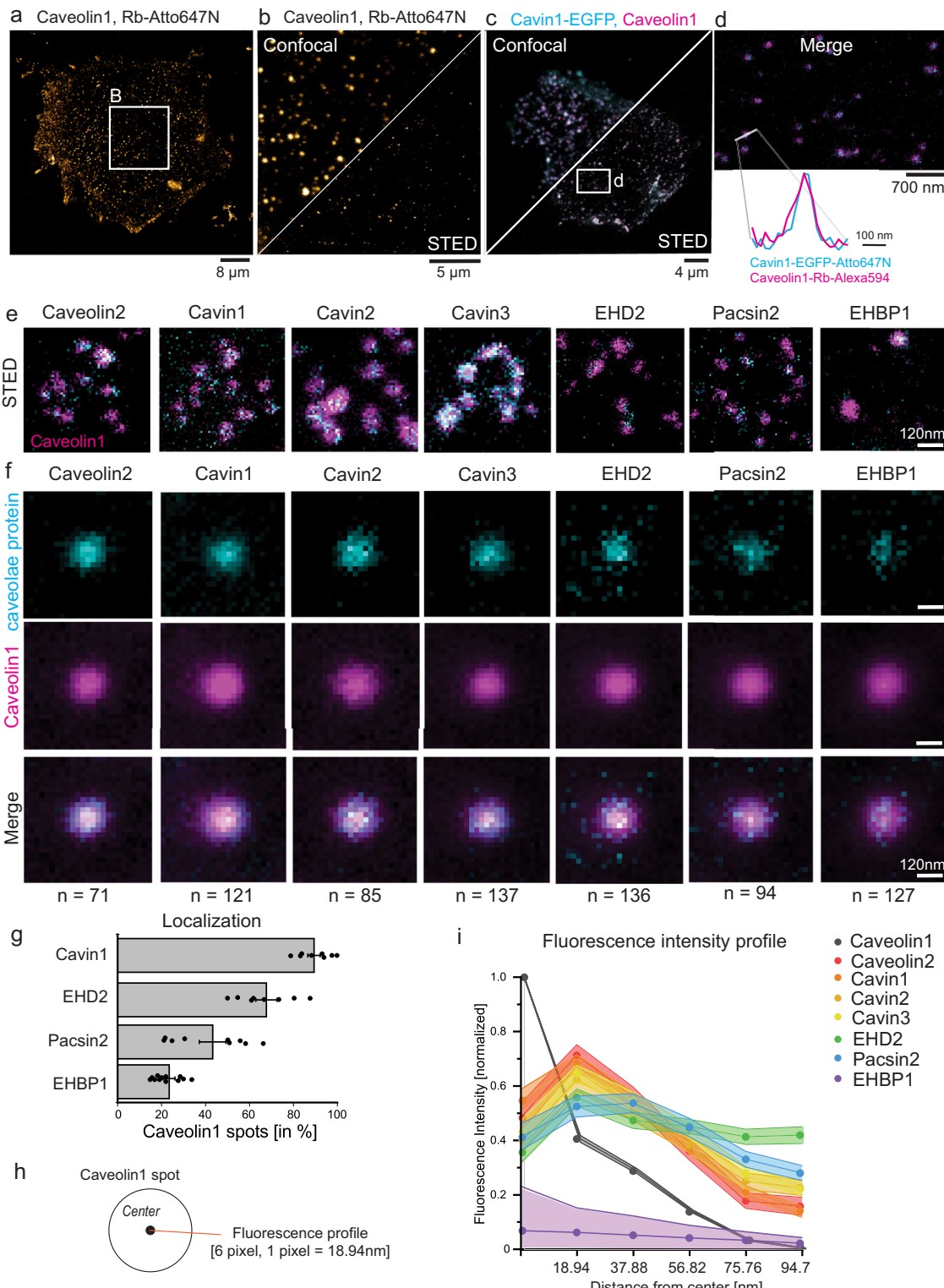

osmotic shock (strong osmotic shock Fig. 5c, d and mild osmotic shock depicted in Suppl. Fig. 7a–c). When cavin1, 2, or 3 were immunolabelled, however, we detected the three isoforms on all caveolae types after both mild (Suppl. Fig. 7) and strong osmotic shock (Fig. 5b, overview images in Suppl. Fig. 8). Quantitative analysis of cavin1-3 staining revealed that all three cavin isoforms were detected at the majority of the low and medium curved caveolae (Fig. 5e). Surprisingly, lower numbers of cavin1-3 positive highly curved caveolae were observed in HUVEC STED-CLEM images

before and after osmotic shock (Fig. 5e). In line with these findings, TIRF microscopy of intact HUVEC cells also revealed strong co-localization of cavin1-3 to caveolin1 at the plasma membrane after both mild and strong osmotic shock (Suppl. Fig. 9). Furthermore, we did not detect measurable cavin1-3 disassembly on caveolae at the plasma membrane after osmotic shock (Suppl. Fig. 9d, e). This is different than current models of cavin behavior, where the cavin coat is proposed to be completely disassembled and lost when caveolae are flat[10,40,49,50].

**Fig. 3 | Stimulated emission depletion microscopy (STED) shows specific protein profiles for caveolae. a** Confocal image of MEF plasma membrane sheet immunolabelled with an antibody against caveolin1 and a secondary anti-rabbit antibody tagged with Atto647N (Rb-Atto647N, 3 independent experiments). **b** Enlarged selection from (**a**) shows confocal and STED image of endogenous caveolin1 in MEFs. **c** Confocal and STED image of cavin1-EGFP expressing MEFs immunolabelled with caveolin1 antibody (secondary antibody dye Alexa594, magenta). Cavin1 was tagged by GFP nanobody labelled with Atto647N (blue, 3 independent experiments). **d** Zoomed STED image section from (**c**). STED fluorescence profile illustrates cavin1 localization to a caveolin1 spot. **e** Representative STED images of caveolae proteins (cyan) and caveolin1 antibody labeling (magenta, secondary antibody tagged with Alexa594) in plasma membrane sheets from MEFs. The individual caveolae proteins were expressed with EGFP tags and labelled with GFP nanobody-Atto647N. **f** Normalized average STED fluorescence intensity projection of automatically detected caveolin1 spots (magenta) and the corresponding co-labeled caveolae proteins (cyan). Lower panel shows both channels as merged image and the total number of caveolin1 spots is indicated. Scale bar represents 120 nm. **g** Percentage of caveolin1 spots that showed localization of either cavin1, EHBP1, EHD2 or pacsin2. Bar plot indicates mean ± SE (n(cavin1) = 1135/8 cells, n(EHBP1) = 1452/11 cells, n(EHD2) = 949/8 cells, n(pacsin2) = 1037/9 cells, 3 independent experiments). **h** The STED fluorescence plot profile for the individual caveolae proteins was analyzed from the center of the caveolin1 spot to the edge. Pixel size in STED images was 18.94 nm, based on the estimated caveolae diameter of 100 nm (radius = 50 nm), the edge of caveolae can be assumed between 3 and 4 pixel from the center (56–75 nm). **i** Fluorescence plot profiles from the center of caveolin1 spots accordingly to (**h**). Line graph indicates mean ± SE for each pixel and caveolae protein (n(caveolin1) = 121, n(caveolin2) = 71, n(cavin1) = 121, n(cavin2) = 85, n(cavin3) = 137, n(EHD2) = 136, n(pacsin2) = 94, n(EHBP1) = 127, 3 independent experiments).

## The caveolae regulatory proteins EHD2 and pacsin2 accumulate at low curved caveolae

Next, we investigated the localization of EHD2 and pacsin2. First, EHD2-EGFP expressing MEFs were imaged by STED-CLEM (Fig. 6a). Previous work has suggested that EHD2 localizes to the neck of caveolae[25]. Correlated STED data, however, showed that EHD2 was also detected at low curved caveolae (Fig. 6a, b). Interestingly, the EHD2 fluorescence signal which is diffuse in less curved caveolae was concentrated around 20 nanometers from the center of highly curved caveolae (as illustrated in the fluorescence plot profile in Fig. 6b). This suggests that EHD2 molecules reposition to the caveola neck during curvature, similar to the dynamics of the related protein dynamin at clathrin-coated pits[51]. Because overexpression of EHD2 results in an increase in curved and immobile caveolae[12,13], we also imaged endogenous EHD2 with antibodies. Figure 6c shows representative antibody-stained CLEM images of low curved caveolae coated with EHD2. Similar to EHD2-EGFP expressing cells, quantitative analysis indicated a distinct signal of EHD2 at both low and highly curved caveolae (Suppl. Fig. 10c).

Only a sub-population (42%) of caveolin1 spots were positive for pacsin2 in the previous STED experiment (Fig. 3g). To identify which caveolae sub-type is enriched in pacsin2, MEFs expressing pacsin2-EGFP were analyzed by STED-CLEM (Fig. 6d, f). Low curved caveolae displayed elevated pacsin2 occupancy compared to highly curved caveolae (Fig. 6g). These observations were verified by antibody staining for endogenous pacsin2, which showed a similar bias to flat structures (Suppl. Fig. 10).

## EHBP1 localizes to highly curved caveolae

The actin and EHD2 binding protein EHBP1 associates with caveolae[17], however its location is unclear. Therefore, STED-CLEM of MEFs overexpressing EHBP1-EGFP was used to image EHBP1. Figure 6e shows representative CLEM images for EHBP1. Quantitative analysis showed that EHBP1 accumulates at highly curved caveolae rather than low curved domains (Fig. 6g). Antibody staining for EHBP1 confirmed this distribution (Suppl. Fig. 10).

In summary, at nascent caveolae domains, cavin proteins accumulate with caveolins when the characteristic well-defined coat is not yet clearly formed. Cavin proteins remain on low curved caveolae formed after osmotic shock. EHD2 is present at all caveolae domains despite the lack of a caveolae neck. Pacsin2 primarily localizes to low curved caveolae, and EHBP1 is enriched in highly curved caveolae.

## Caveolae regulatory proteins modulate curvature

Next, we tested how reduced levels of EHD2, pacsin2, or EHBP1 effect caveolae shape and density (Fig. 7a, Western Blot verification Suppl. Fig. 11a–c). MEFs lacking EHD2[24] exhibited no change in the number of caveolae (Fig. 7a, b), however substantially more highly curved caveolae were observed compared to wild-type MEFs (Fig. 7d). siRNA smart-pool based knockdown of pacsin2 resulted in an increase in lower curved caveolae (40% vs. 24% in wild-type), while reducing the percentage of highly curved caveolae (14% from 25% in wild-type, Fig. 7a–d). Knockdown of EHBP1 by a pool of 4 siRNAs did not alter caveolae number or proportion of types (Fig. 7d). Interestingly, the loss of all three proteins (double knockdown of Pacsin2 and EHBP1 in EHD2 lacking cells, triple KO/KD, see also Western Blot in Suppl. Fig. 11a–c) substantially reduced the total number of caveolae at the plasma membrane (Fig. 7c). In particular, low and medium curved caveolar invagination percentages were reduced in this condition compared to wild-type cells. Surprisingly, the triple KO/KD MEFs showed more highly curved caveolae in comparison to wild-type MEFs (Fig. 7d). 53% of all caveolae were highly curved in triple KO/KD MEFs compared to 25% in wild-type MEFs. Notably, caveolin1 and cavin1 protein levels were also reduced in triple KO/KD MEFs (Western Blot Suppl. Fig. 11d). Size measurements for the individual caveolae showed that the loss of EHD2, pacsin2, or EHBP1 results in slightly smaller caveolae with the exception of low curved caveolae in Pacsin2 knockdown MEFs (Fig. 7e).

## Dynamin does not localize to caveolae

Dynamin has been implicated in caveolae endocytosis[26]. Caveolae mobility and endocytosis are inhibited in cells expressing the dynamin mutant K44A (Dyn-K44A)[24,30,52]. However, previous studies largely failed to strongly localize dynamin at caveolae. Therefore, we mapped dynamin at individual caveolae. First, STED was used to detect dynamin2-EGFP (the major isoform in MEFs[53]) at the plasma membrane (Fig. 8a). Surprisingly, no substantial co-localization of dynamin2 (Fig. 8a, cyan) and caveolin1 (Fig. 8a, magenta) was observed. Yet, dynamin2 was commonly and strongly detected at clathrin sites (Fig. 8a, yellow). The Dyn-K44A mutant has been proposed to accumulate at caveola necks[14,29,54]. In contrast, we rarely observed co-localization of Dyn-K44A and caveolin1 (Fig. 8a, lower panel). Quantitative analysis of the fluorescence signals showed that dynamin2 or Dyn-K44A localized to only 8.9 ± 1.1% or 18.2 ± 2.4% of caveolae. Dynamin mutant showed a modest increase over background (Fig. 8b, c). Thus, Dyn-K44A was used to study dynamin in STED-CLEM (Fig. 8d). As expected, dynamin was strongly localized to clathrin-coated sites (Fig. 8d -II). However, dynamin-K44A did not strongly localize with caveolae in these images (Fig. 8d · I). To complement these observations from transfected cells, MEFs were immuno-stained against endogenous dynamin. Again, no substantial localization of dynamin at caveolae was detected, also when MEFs were treated with oleic acid to induce caveolae endocytosis (Suppl. Fig. 12). Furthermore, STORM-CLEM of dynamin2 in HeLa and SK-MEL-2 cells[46] did not show noticeable association of dynamin with caveolae (Suppl. Fig. 13, dynamin-GFP and endogenous dynamin). In addition, Dyn-K44A STORM-CLEM in SK-MEL-2 cells did not show robust accumulation at caveolae (Suppl. Fig. 13e, f). The common close positing of clathrin and

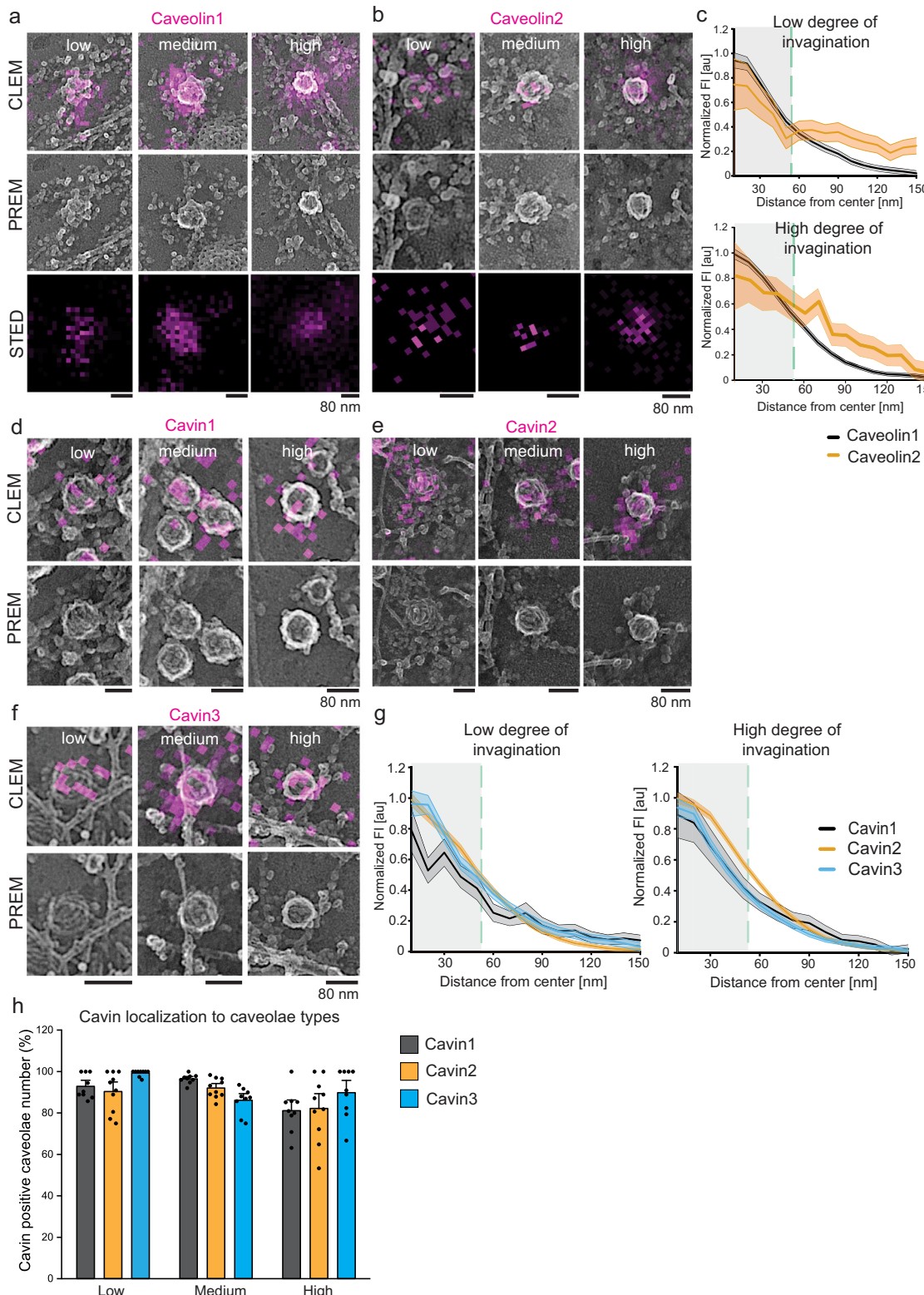

caveolae at the plasma membrane—where on occasion dynamin localizes between the two organelles—might lead to false-positive co-localizations between caveolae and dynamin in fluorescence images (Suppl. Fig. 13b). This may explain past suggestions of co-localization. These false positives would be invisible without CLEM or three color super-resolution imaging of dynamin, clathrin, and caveolae.

To test for a functional role for dynamin in caveolae curvature, we examined if there was a morphological change to caveolae when dynamin was absent. Specifically, we investigated caveolae number and shape in MEFs lacking all three dynamins[55]. If dynamin is involved in caveolae membrane scission, loss of dynamin should increase caveolae number. PREM of dynamin triple knockout cells (dynamin1/2/3 knockout[55], Suppl. Fig. 14 Western Blot validation) showed no

**Fig. 4 | STED-CLEM revealed the localization of cavin proteins to low, medium and highly curved caveolae. a–b** Representative CLEM images for MEFs expressing either caveolin1-EGFP (**a**) or caveolin2-EGFP (**b**) which were labelled with GFP nanobody-Atto647N and imaged with STED and PREM. **c** Quantitative analysis of low and highly curved caveolae STED fluorescence profiles from the center of caveolae to their edges (indicated by green dashed line, line graphs show mean ± SE; caveolae number: caveolin1: n(low degree of invagination) = 145, n(high degree of invagination) = 178; caveolin2: n(low) = 140, n(high) = 474, 4 independent experiments). **d–f** Representative CLEM images for MEFs expressing either cavin1-EGFP (**d**), cavin2-EGFP (**e**), or cavin3-EGFP (**f**) labelled with GFP nanobody-Atto647N and investigated by STED followed by PREM. **g** Quantitative analysis of low and highly curved caveolae by STED fluorescence profiles from the center of caveolae to their edges (indicated by green dashed line; line graphs show mean ± SE; caveolae number: cavin1: n(low) = 59, n(high) = 272; cavin2: n(low) = 231, n(high) = 223; cavin3: n(low) = 198, n(high) = 191, 4 independent experiments). **h** Quantitative analysis of cavin1-3 localization to different caveolae types. Bar graph indicates caveolae stained positive for cavin1, 2 or 3 related to all caveolae per curvature type detected in CLEM images. Bar graph indicates mean ± SE; caveolae number: cavin1: $n$ = 821/9 cell regions; cavin2: $n$ = 1064/10 cell regions; cavin3: $n$ = 480/9 cell regions, 4 independent experiments).

substantial changes in caveolae shape or density (Fig. 8e). Quantitative analysis of caveolae number at the plasma membrane did not reveal a significant difference compared to wild-type cells (Fig. 8f). Furthermore, the percentage of low, medium, and highly curved caveolae was unchanged in dynamin-lacking cells (Fig. 8g). Highly curved caveolae showed a slight reduction in size compared to wild-type MEFs (Fig. 8h). The previously-observed size decrease from low to highly curved caveolae (Fig. 2e) was also detected. In summary, we find no strong evidence that dynamin localizes to caveolae or has a mechanistic role in these organelles at the plasma membrane. These data indicate that dynamin's previously-reported effects on caveolae could be indirect, very transient, or occur on only a small number of caveolae. Furthermore, dynamin could impact caveolae through indirect effects on clathrin-mediated endocytosis, general membrane traffic, membrane tension, or the actin cytoskeleton. Future work is needed.

## Discussion

Caveolae are one of the most common organelles found at the plasma membrane of many human cell types. It is still unclear, however, how caveolae assemble, change their curvature, and are captured into the cell. We analyzed the nanoscale architecture of caveolae marked with key proteins. We classified caveolae into three sub-types according to curvature: low, medium, and highly curved invagination. In the different cell lines the percentage of low, medium and highly curved caveolae was similar. The majority of caveolae are medium curved (bulb-shape, 55–62%). A smaller number contain only low curvature (11–18%). Likely, prior to endocytosis, caveolae transition into highly curved invaginations (20–34%), forming the constricted neck needed for membrane scission. These highly curved caveolae might spend a short amount of time in this state at the plasma membrane before transport into the cytosol. The smaller number of highly curved caveolae compared to low or medium curved caveolae may reflect the reported mobile endocytic caveolae[56]. Interestingly, PREM images still revealed more highly curved caveolae, which is surprising given the rather low rates of endocytosis proposed for caveolae[56–58]. This discrepancy may be explained by the fact that the PREM images mainly evaluate caveolae in fixed cells and cannot identify caveolae that would be seen as highly mobile in live cell TIRF and uptake measurements. Furthermore, cell types may differ in their caveolae endocytosis rates.

We measured a reduction in the radii of caveolae from low to highly curved caveolae curvature. A reduced radii in highly curved caveolae has also been observed after extracellular lipid treatment[59]. While the radii decreased, the calculated surface area of individual caveolae increased as caveolae curve, indicating that bending captures excess plasma membrane. Thus, during caveolae curvature, a cell will capture and reduce the exposed plasma membrane surface. Indeed, previous data showed that lipid accumulation in caveolae domains can prime invaginations for endocytosis[56,57,59]. Possibly, new lipids are required for this process[2,36]. This could be an important mechanism for lipid uptake.

How do proteins drive membrane curvature at caveolae? First, caveolae, regardless of curvature, contained three major proteins (caveolin, cavin, EHD2). Unlike past models, cavins1-3 were found at low curved caveolae. This was also true after treating cells with a mild or strong osmotic shock to flatten caveolae. Here, all three cavin isoforms remain associated with caveolae. In past studies, cavins have been proposed to localize only to strongly curved caveolae and were lost when caveolae flatten[3,10,40,49]. The loss of cavin was proposed to drive flattening and thus lower caveolae curvature. Previous studies focused on cavin localization to caveolin1 but were unable to ascertain the curvature due to a lack of correlative fluorescence and electron microscopy. Therefore, these studies failed to evaluate membrane curvature in caveolae domains after osmotic shock treatment. In this study, the replica membrane sheets showed that caveolae containing the cavin coat can flatten. Similar results were observed after mild osmotic shock treatment when caveolae were inspected with STORM super-resolution microscopy[60]. Notably, our STED-CLEM approach cannot evaluate quantitative changes in protein levels at caveolae membrane sites. Consequently, different results will be observed compared to other analysis of cavin plasma membrane localization and cavin-caveolin1[40,49,61] or caveolin1-caveolin1[62,63] association after osmotic shock. Future experiments are needed to dismantle these discrepancies. Previous studies reported that methyl-beta-cyclodextrin treatment which removes cholesterol from the plasma membrane resulted in more caveolae with lower curvature[47,64]. The combination of cholesterol removal and CLEM could be a helpful approach to answer these questions.

Second, we find that EHD2 localizes to caveolae independent of the underlying curvature. EHD2 has been proposed to associate with only the neck of caveolae[13,25]. Here, we observed both endogenous EHD2 or expressed EHD2-EGFP at both low and highly curved caveolae. When comparing STED images of both caveolae types, EHD2 appears slightly more diffuse around low curved caveolae (Fig. 6). Structural data of EHD2 and related EHD proteins (such as EHD4) demonstrated an ATP-dependent oligomerisation[65–68] at lipid bilayers that forms a ring. Yet, it was recently shown that EHD can bind to flat membranes as a filament. These filaments change conformations to induce tubulation[67]. A more constricted EHD4 filament was observed when the underlying membrane curvature was increased. Therefore, we speculate that the accumulation of EHD2 around low curved caveolae could be a nucleus or tether for its subsequent ring-like polymerization at the caveolar neck. Together with pacsin2, which is also frequently detected at low curved caveolae, increased membrane curvature could be generated. Notably, highly curved caveolae showed a more confined EHD2 localization that suggests a dense ring-like structure near the caveolar neck. More detailed 3D atomic data is needed to fully understand how EHD2 oligomers (in concert with pacsin2) form at caveolae. As EHD2 stabilizes caveolae at the plasma membrane, the loss of EHD2 results in highly mobile caveolae[12,13,24] which is reflected in an increase of highly curved caveolae. Of note, similar to our observations, EHD2 deletion in vivo did not alter caveolae number in some organs[24,69].

In summary, from these data we conclude that low and highly curved caveolae have similar core protein profiles which include caveolin, cavins, and EHD2, that are flexible polymers which can accommodate a range of curvatures and remain associated with each

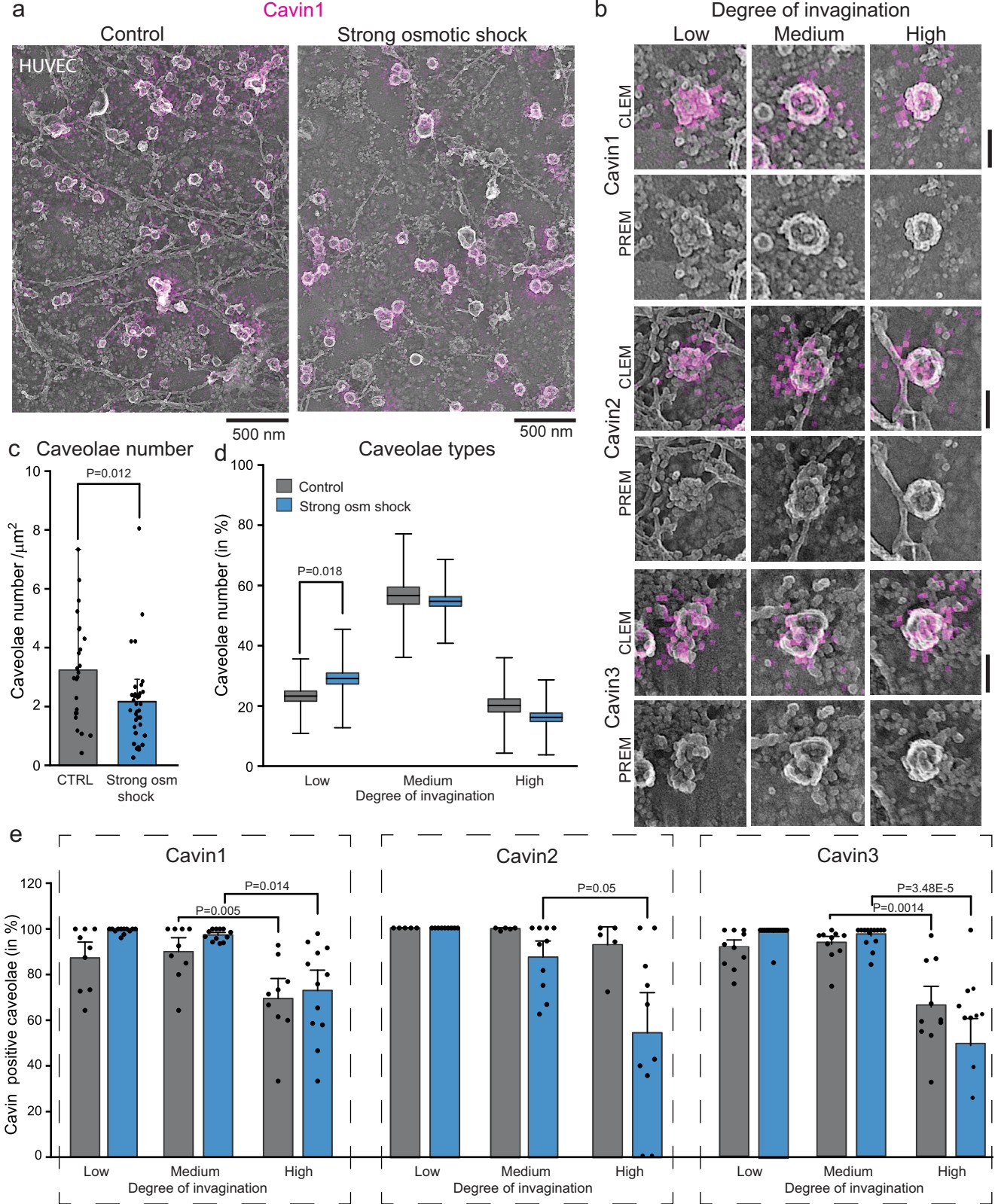

other and the membrane during these transitions (model Fig. 9). This indicates that changes in membrane curvature at caveolae coats can occur without disassembly, re-assembly, or major re-organization of the coat. Possibly, lipid changes drive caveolae curvature changes as previously suggested[2,3,59,70,71].

We also investigated the localization of the relatively under-studied protein EHBP1 at caveolae. EHBP1 is known to bind to EHD2 proteins and actin[72–74], and has been suggested to be involved in caveolae related processes[17,74]. EHBP1 was localized to only a subset of caveolae (Fig. 3). Morphological analysis revealed that it is more likely associated with highly curved caveolae. However, loss of EHBP1 did not alter caveolae number or shape (Fig. 7) indicating a likely regulatory rather than structural role of this protein.

**Fig. 5 | Preserved cavin localization to low curved caveolae after osmotic shock in HUVEC. a** Representative STED-CLEM image of endogenous cavin1 antibody staining in HUVEC membrane sheets treated with strong osmotic shock (1:9 dilution with deionized water). **b** Representative STED-CLEM images of low, medium and highly curved caveolae endogenously stained against cavin1, 2 or 3 after strong osmotic shock in HUVEC. Scale bar is 80 nm. **c** Total caveolae number per membrane area in untreated or strong osmotic shock treated HUVEC (n(control) = 24 cell regions, n(strong osmotic shock) = 36 cell regions, 5 independent experiments). **d** Caveolae numbers (in %) of low, medium or highly curved caveolae in control and strong osmotic shock treated HUVEC (control: n(low) = 647, n(medium) = 2053, n(high) = 553; strong osmotic shock: n(low) = 918, n(medium) = 1975,

n(high) = 554, 5 independent experiments) (**e**) Cavin1-3 localization to different caveolae types. Bar graph indicates caveolae stained positive for cavin1, 2 or 3 related to all caveolae per curvature type detected in CLEM images (in %, caveolae number: cavin1: n(control) = 407/9 cell regions, n(strong osmotic shock) = 1691/12 cell regions; cavin2: n(control) = 884/5 cell regions, n(strong osmotic shock) = 381/10 cell regions; cavin3: n(control) = 1962/10 cell regions, n(strong osmotic shock) = 1375/14 cell regions; 5 independent experiments). Bar and box graph indicate mean values ± SE, whiskers show SD, each replicate is depicted. Normal distributed groups were analyzed by two-sided *t*-test, not normally distributed values with two-sided Mann–Whitney test.

The proteins pacsin2 and EHBP1 were observed with more dispersed and sporadic localization profiles. Pacsin2 was mainly found at lower curved caveolae, where EHBP1 preferred highly curved invaginations. This suggests that both proteins might be dynamically involved in the regulation of caveolae localization and traffic. Indeed, in the absence of pacsin2, caveolae appeared less curved (Fig. 7). In line with published results[14–16,27] this indicates that pacsin2 is likely involved in caveolae curvature generation. Surprisingly, loss of pacsin2 and EHD2 combined (independently of EHBP1 levels) changed this result (Fig. 7) as the majority of caveolae became highly curved. Furthermore, less caveolae were observed at the plasma membrane. Previous studies showed that caveolins and cavin proteins alone are able to form heterologous caveolae[19,75–77] and, importantly, cellular uptake can occur[75]. Therefore, we speculate that the loss of EHD2, pacsin2, and EHBP1 leads to a unique caveolae structure. These "minimal" caveolae (containing caveolae coat proteins only) are much more mobile and less stable at the plasma membrane. This supports the idea that EHD2 restrains caveolae at the plasma membrane and pacsin2 is important for highly curvature formation.

Dynamin has been proposed to facilitate caveolae capture from the plasma membrane[26]. Surprisingly, across multiple experimental systems, we failed to clearly localize dynamin to caveolae (Fig. 8). In addition, loss of all endogenous dynamins did not alter caveolae number or curvature. Thus, a direct physical role for dynamin at caveolae is not supported by our data. However, we cannot exclude that specific cellular triggers could induce dynamin accumulation. Likewise, a direct binding of dynamin to caveolae could be very transient or very sparse. These would be difficult to detect with our imaging methods. In contrast, dynamin was strongly localized at nearby clathrin coated structures in abundance. How is caveolae fission achieved? Recently, Larsson et al. reported that dynamin loss increased caveolae mobility which may suggest that other currently unknown mechanisms play a role in this process[78]. Besides its well-studied function in clathrin mediated endocytosis, dynamin can interact with actin[79–81]. Dynamin GTPase mutant K44A inhibits actin dynamics[81]. Caveolae are able to bind actin[82] and when expressing the dynamin K44A mutant, caveolae mobility and endocytosis is inhibited[24,29,30,52]. Thus, we suggest that dynamin may be involved in actin-dependent caveolae traffic[82]. Possibly the combined functions of EHD2, pacsin2 and EHBP1 shapes and stabilizes the caveolar neck. Removal of these regulatory proteins shifts invaginated caveolae toward more highly curved spheres (as shown in triple KO/KD MEFs, Fig. 6). Binding of actin filaments to cavins or caveolins (via linker such as filamin A[82]) may then introduce additional mechanical force needed to overcome the energy barrier preventing membrane fission and endocytosis. Here, dynamin may form actin bundles with enhanced mechanical strength that allow the pulling of caveolae from the plasma membrane in a manner similar to membrane protrusions during cell fusion[79] or clathrin-independent endocytosis[83]. Likewise, a general role of dynamin on actin-based membrane tension or cortical cytoskeleton organization could impact caveolae behaviors. Future work is needed to clearly determine dynamin's exact role in caveolae dynamics.

There are several specific limitations to our study. First, we focus on caveolae at the bottom (ventral) membrane of single cells. This is also true for most studies of caveolae that rely on evanescent field microscopy. Whether caveolae at the top membrane show similar protein composition and structural features will need to be determined. A similar issue exists for cells in complex tissues where they might be contacting other cells in three dimensions. Second, the resolution for STED of 40–60 nm is still too large to resolve subtle sub-caveolae localization differences between the caveolae-associated proteins. If there is a slight heterogeneity of the eight proteins studied here, it will require much higher-resolution imaging methods such very high resolution light microscopy, expansion microscopy, or cryotomography at the atomic scale. Third, slight shifting of the fluorescence signal compared to the underlying membrane can occur during preparations steps between light and EM yet these effects are small as we have shown in past work[46,84,85]. The majority of correlated caveolae fluorescence match the underlying membrane shapes in their corresponding replica plasma membrane sheets. Fourth, we removed the unbound cytosol and nucleus with unroofing. While past studies have shown that this does not substantially change the plasma membrane or organelle structures in our systems, any subtle alterations to the underlying organelles will require higher-resolution and faster imaging studies in intact living cells. Yet, as we see unexpected association (and not loss) of proteins at caveolae, we believe that the possible perturbations induced by unroofing do not impact our major experimental findings.

## Methods
### Cell culture
Wildtype, EHD2 knockout mouse embryonic fibroblasts (MEFs, previously described[24]), 3T3-L1 fibroblasts (Atcc Cat# CL-173) and dynamin knockout MEFs (generously shared by Pietro De Camilli[55,86]), were cultured in Dulbecco's modified Eagle's medium (DMEM, Gibco #11965092) supplemented with 10% fetal bovine serum (FBS, Atlanta Biologicals #S10350) and 1% penicillin and streptomycin (Gibco #15140122). HUVEC (Promocell #C-12203) were cultured in endothelial cell growth basal medium including SupplementMix (Promocell #C22010). Medium was changed every 2 days. For fluorescence or EM experiments cells were seeded on fibronectin coated glass dishes (#1.5 high precision, 25 mm; for STED: etched grid coverslip Bellco Biotechnology #1916-91012) and cultivated for 24–48 h at 37 C in 5% CO₂. MEFs were used for experiments until passage 35, HUVEC were used until passage 7. Triple dynamin (Dyn1/2/3) knockout was induced by 1 μM 4-hydroxytamoxifen (Sigma #94873) as previously described[55]. Briefly, MEFs were seeded sub-confluent and 1 μM 4-hydroxytamoxifen for 2 days was applied, followed by fresh DMEM containing 300 nM 4-hydroxytamoxifen for 4 days. Dynamin protein level was evaluated after 6 days by Western Blot and experiments were performed.

### Hypo-osmotic shock in HUVEC
Osmotic shock was induced by incubation of HUVEC in pre-warmed 1:5 (mild) or 1:9 (strong, v%) growth medium diluted with deionized water

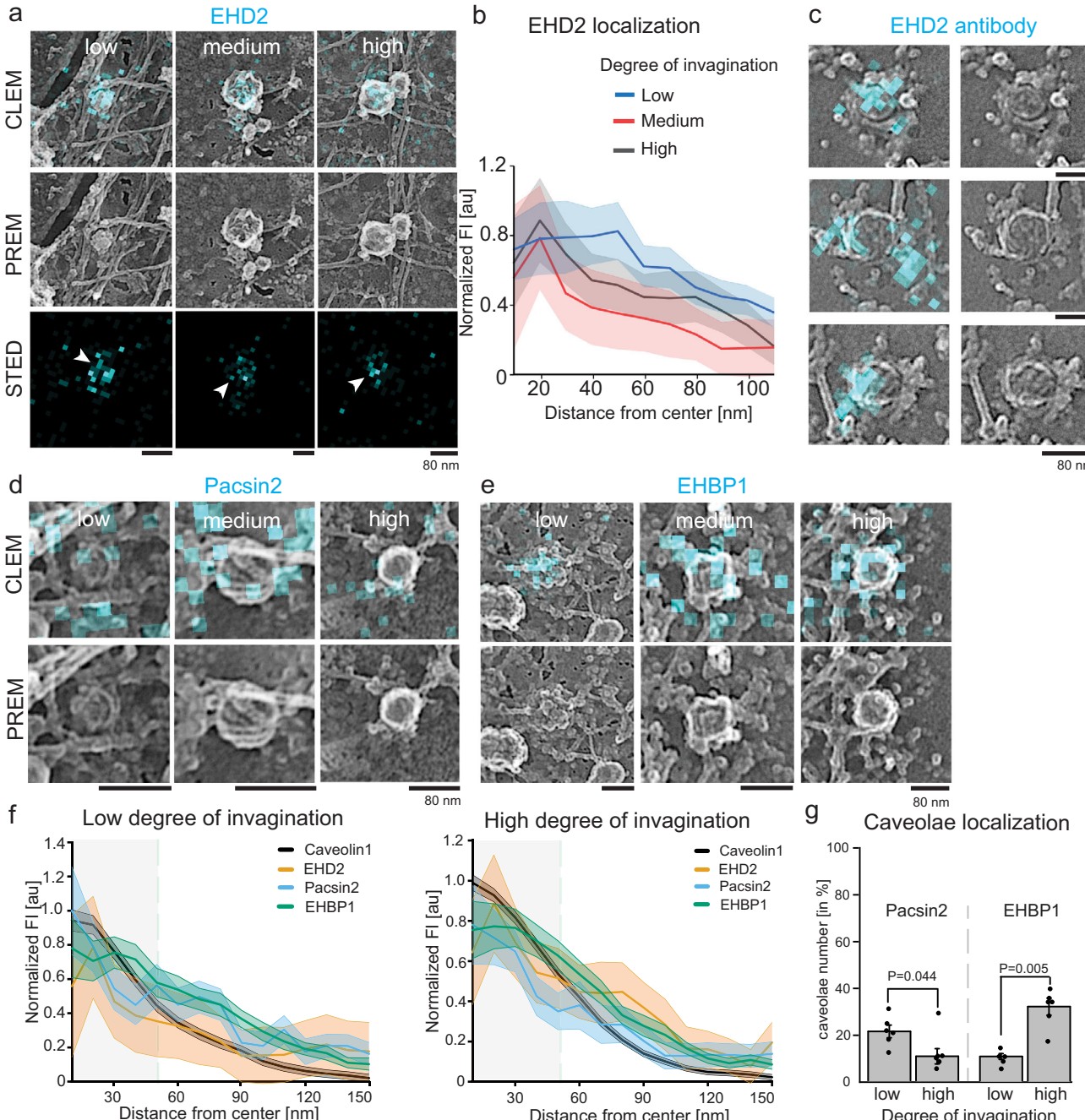

**Fig. 6 | Spatially distribution of EHD2, pacsin2 and EHBP1. a** Representative STED-CLEM images for MEFs expressing EHD2-EGFP (labelled with GFP nanobody-Atto647N) or white arrows indicate accumulation of EHD2 around the caveolae center. **b** EHD2 STED fluorescence profile from the center of caveolae to its edges obtained from STED-CLEM images (**a**). Each individual caveolae type is depicted (graph shows mean ± SE, caveolae number: n(low) = 72, n(medium) = 50, n(high) = 163, 2 independent experiments). **c** STED-CLEM showing endogenous EHD2 antibody staining at low curved caveolae (secondary antibody tagged with Atto647N, 2 independent experiments). **d,e** Representative CLEM images for MEFs expressing either pacsin2-EGFP (**d**) or EHBP1-EGFP (**e**) that were labelled with GFP nanobody-

Atto647N (cyan). **f** Quantitative analysis of low or highly curved caveolae by STED fluorescence profiles from the center of caveolae to their edges (indicated by green dashed line, line graphs show mean ± SE, caveolae number: caveolin1: n(low) = 145, n(high) = 178; EHD2: n(low) = 72, n(high) = 163; pacsin2: n(low) = 103, n(high) = 77; EHBP1: n(low) = 70, n(high) = 79, 3 independent experiments). **g** Percentage of low or highly curved caveolae that were targeted by either pacsin2 or EHBP1. Bar plot indicates mean ± SE, n(pacsin2) = 1357 caveolae/3 cells, n(EHBP1) = 660 caveolae/4 cells, 3 independent experiments. Significant difference was tested by two-sided Mann–Whitney test. Scale bar is 100 nm.

for 5 min[40]. Afterwards HUVEC were immediately unroofed and fixed to prepare plasma membrane sheets. Evaluation of osmotic shock was done by phase contrast imaging inspecting cell shape over the time-course of 1–10 min.

### Plasmid transfection and siRNA treatment
Lipofectamine3000 (Invitrogen #L3000015) was used to transfect MEFs seeded in 6 well plates (100.000 cells/well) with 2.5 µg plasmid accordingly to the manufacturers protocol. siRNA treatment

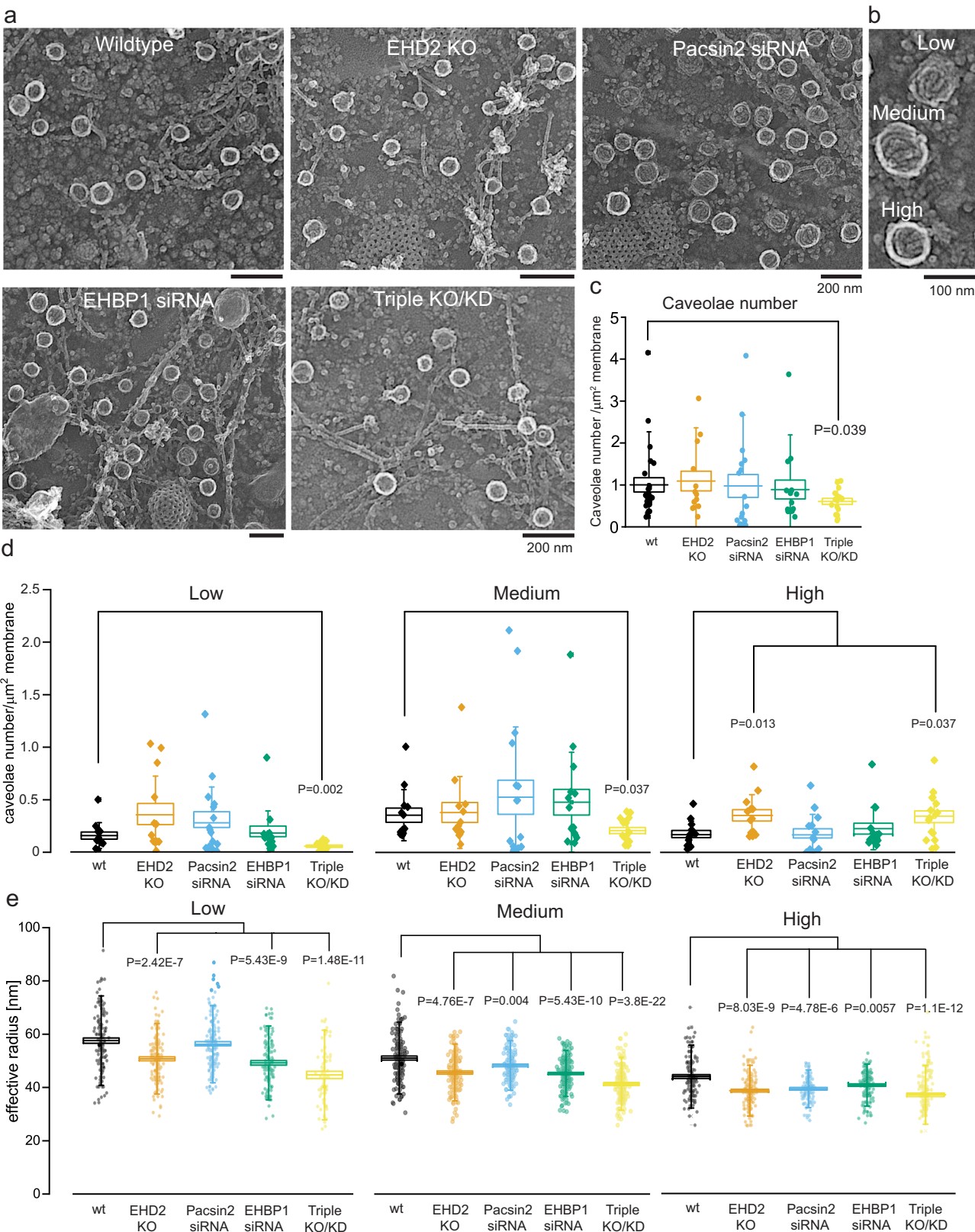

was performed with Lipofectamine RNAiMax (Invitrogen #13778150), whereby the final siRNA concentration was 50 pmol per well (6 well plate). SMARTpool (mix of 4 siRNAs/target, Dharmacon) against mouse pacsin2 (#M-045093-01-0005) and mouse EHBP1 (#M-052068-01-0005) were used to obtain sufficient knockdown

which was evaluated by Western blotting. All experiments were carried out after 48 h incubation. The following plasmids were used: pCaveolin1-EGFP, pCaveolin2-EGFP, pCavin1-EGFP, pCavin2-EGFP, pCavin3-EGFP, pPacsin2-EGFP, pEHD2-EGFP, pEHBP1-EGFP, pHis-Cavin1-EGFP, pDynamin2-GFP, pDynamin2-K44A-GFP.

**Fig. 7 | EHD2, pacsin2 and EHBP1 stabilize caveolae at the plasma membrane.**
**a** Representative PREM example images of wildtype MEFs (wt), EHD2 knockout
MEFs (KO), pacsin2 siRNA or EHBP1 siRNA treated MEFs. Triple knockout/knock-
down (KO/KD) indicates EHD2 KO MEFs that were treated with pacsin2 and
EHBP1 siRNA. Scale bar is 200 nm. **b** Example PREM image of low, medium and
highly curved caveolae. **c** In PREM images the total caveolae number at the plasma
membrane in wt, EHD2 KO, pacsin2 siRNA treated, EHBP1 siRNA treated, or triple
KO/KD MEFs was measured (n(wt) = 25 cell regions, n(EHD2 KO) = 13 cell regions,
n(pacsin2 siRNA) = 16 cell regions, n(EHBP1 siRNA) = 15 cell regions, n(Triple KO/
KD) = 16 cell regions, 3 independent experiments). **d** Number of individual caveolae
types in wt, EHD2 KO, pacsin2 siRNA, EHBP1 siRNA or triple KO/KOD MEFs
(n(wt) = 13 cell regions, n(EHD2 KO) = 13 cell regions, n(pacsin2 siRNA) = 17 cell

regions, n(EHBP1 siRNA) = 15 cell regions, n(Triple KO/KD) = 16 cell regions, 3
independent experiments). **e** Caveolae radius (round caveolae domains were
assumed) of low, medium and highly curved caveolae in wt, EHD2 KO,
pacsin2 siRNA, EHBP1 siRNA or triple KO/KOD MEFs (caveolae number: low:
n(wt) = 127, n(EHD2 KO) = 139, n(pacsin2 siRNA) = 163, n(EHBP1 siRNA) = 123,
n(Triple KO/KD) = 69; medium: n(wt) = 148, n(EHD2 KO) = 130,
n(pacsin2 siRNA) = 146, n(EHBP1 siRNA) = 177, n(Triple KO/KD) = 145; high: n(wt) =
129, n(EHD2 KO) = 157, n(pacsin2 siRNA) = 106, n(EHBP1 siRNA) = 142, n(Triple KO/
KD) = 133, 3 independent experiments). Box plots indicate mean values ± SE,
whiskers show SD, each replicate is depicted, statistical significance was measured
by two-sided *t* test in normally distributed data sets, otherwise the nonparametric
two-sided Mann–Whitney test was applied.

## Preparation of plasma membrane sheets

Cells were unroofed prior to immunofluorescence staining or TEM
preparation to obtain plasma membrane sheets as described pre-
viously ([45,85]). Briefly, cells seeded on glass dishes were washed with PBS
and cell membrane stabilization buffer (70 mM KCl, 30 mM HEPES
maintained at pH 7.4 with KOH, 5 mM MgCl$_2$, 3 mM EGTA), and placed
in fresh stabilization buffer. The unroofing was performed with 2% PFA
(EM grade, freshly prepared, Electron Microscopy Science #15710) that
was splattered with a 19-gauge needle and syringe on the cells. After-
wards, the unroofed cells were placed in fresh 4% PFA (for immuno-
fluorescence) or in 2% glutaraldehyde (for TEM, EM grade, Electron
Microscopy Science #16019) for fixation at 4 C.

## Immunofluorescence staining and dyes

The unroofed cells were incubated in 3% bovine serum albumin/PBS
(BSA, m/v, fresh, Fisher Bioreagents #BP9703) for 1.5 h, followed by
primary antibody (1:100 in 3%BSA/PBS) incubation for 1 h. Next, cells
were washed thoroughly with PBS and the secondary antibody tagged
with fluorescence dye (1:500) or GFP-nanobody (1:500) was applied for
1 h. Afterwards, cells were washed 4 times in PBS and stored in fresh
PBS at 4 C until the samples were imaged. The following antibodies
were used: anti-Caveolin1-Rabbit (abcam #ab2910), anti-Caveolin1-
mouse (Santa Cruz #sc-53564), anti-Cavin1-Rabbit (abcam #76919),
anti-Cavin2-Rabbit (abcam #ab76867), anti-Cavin3-Rabbit (abcam
#abcam2912), anti-EHD2-goat (abcam #ab23935), anti-Pacsin2-Rabbit
(Proteintech #10518-2-AP), anti-EHBP1-Rabbit (Proteintech #17637-1-
AP), anti-mouse-Clathrin heavy chain (Thermo-Fisher #MA1-065,
1:2000), anti-mouse-Dynamin2 (Santa Cruz, C-18; #sc-6400), anti-
rabbit-Atto647N (Rockland #611-156-122), anti-goat-Atto647N (Rock-
land, #610-156-121), anti-goat-Atto647N (Rockland #605-456-013 S),
anti-rabbit-Alexa568 (Invitrogen #A11036), Fab2-anti-rabbit-Alexa594
(ThermoFisher #A-11072), Fab2-anti-mouse-Alexa488 (ThermoFisher
#A-11017), Fab2-anti-mouse-Alexa568 (ThermoFisher #A-11019), GFP-
nanobody-Atto647N (Chromotek #gba647n-100), Phalloidin-Alexa488
(ThermoFisher #A12379).

## STED microscopy

Leica TCS SP8 microscope was used for 3 color gated STED with 100×
objective (NA), including tunable white laser 470–670 nm, 775 and
592 nm depletion laser, and PMT and HyD Sp GaAsP detectors. The
stained (unroofed) cells were imaged in PBS at room temperature.
Depletion laser levels for Atto647N was between 25 and 50%, for
Alexa594 between 40 and 75%, whereby caveolin1 spot diameter size
was used for STED evaluation. STED image size was 19.394 μm with a
pixel size of 18.94 nm. Final lateral resolution was between 40 and
60 nm as determined with 40 nm fluorescent beads (Abberior).

## TIRF microscopy

Intact HUVEC cells were treated either with mild (1:5) or strong (1:9 v/v
medium diluted with water) osmotic shock for 5 mins, and fixed with
4% PFA for 10 min. After washing with PBS cells were permeabilized
with 3%BSA/PBS/0.01% Tween20 for 20 min, followed by 30 min 3%

BSA/PBS blocking. Endogenously antibody staining was performed as
described above. TIRF imaging was performed in PBS, on a Nikon
NSTORM system equipped with an Andor iXon Ultra 897 EMCCD (15.6
photoelectrons per A/D count, 160 nm pixels with ×100 objective, 100
gain). Cavin1 and caveolin1 co-localization was evaluated in ImageJ by
measuring the Pearson correlation with Coloc2.

## Gold labeling of cavin1 and caveolin1

Membrane sheets were prepared as described above, followed by
fixation with 4% PFA for 20 min. Afterwards, the membranes were
washed extensively with PBS (4–5× times), followed by two 0.1% EDTA/
PBS (Sigma #03609) washing steps and incubation with 3% BSA/PBS
(m/v, fresh) for 1 h. 10 nm Ni-NTA-Nanogold (Nanoprobes #2084)
solution was diluted 1:5 in PBS and added to His-Cavin1-EGFP over-
expressing MEF membrane sheets. The samples were first incubated
for 15 min on orbital shaker followed by 45 min incubation without
shaking. Next, the cells were treated similarly to Platinum replica as
described below[84]. Caveolin1 was tagged with specific antibody (anti-
Caveolin1-Rabbit, abcam #ab2910, 1:100 in 3% BSA/PBS), and a sec-
ondary Rabbit antibody labelled with 12 nm gold particles (Dianova
#111-205-144, 1:30 in 3% BSA/PBS) was applied. Investigation of gold
labeling on Pt replica membrane sheets was done by TEM.

## Platinum replica preparation

The plasma membrane sheets were prepared for TEM as described
previously[45]. Briefly, the unroofed cells were fixed in 2% glutaraldehyde
for at least 20 min, followed by extensive washing with PBS and tannic
acid (1 mg/ml dest. H$_2$O) treatment for 20 min. Next, the cells were
stained with 0.1% (v/v) uranyl acetate for 20 min. Afterwards, the
membrane sheets were dehydrated by an increasing EtOH row
(15–100%), followed by critical point drying with CO$_2$. Platinum and
carbon coating was carried out in Leica ACE900 freeze fracture in
which, at first, 3 nm Pt and secondly 5.5 nm carbon was applied on the
membrane sheets. The glass dish of the coated samples were removed
by 10% (v/v) hydrofluoric acid, and the replicas were placed on For-
mvar/carbon coated 75 mesh EM copper grids (Ted Pella #01802-F).

## TEM

TEM imaging was performed using a JOEL 1400 microscope at 15,000
magnification (Pixel size 1.23 nm). Electron tomograms were obtained
at 12,000 magnification (Pixel size 1.56 nm) from −60 to 60 degrees,
with 1 degree increment. To obtain montage TEM images and tomo-
grams SerialEM software was used[87]. Etomo/3DMOD was used to align
tomogram stacks in fiducial-less mode with patch tracking, and IMOD
was used for analysis[88].

## STED-CLEM

Correlation of STED and TEM images was achieved by using gridded
glass coverslips for correct cell assignment. After STED imaging a
confocal tile scan of the grid was acquired including the cells of
interest, followed by replica EM processing as described above. The
region of interest was then cut out from the glass grid and phase

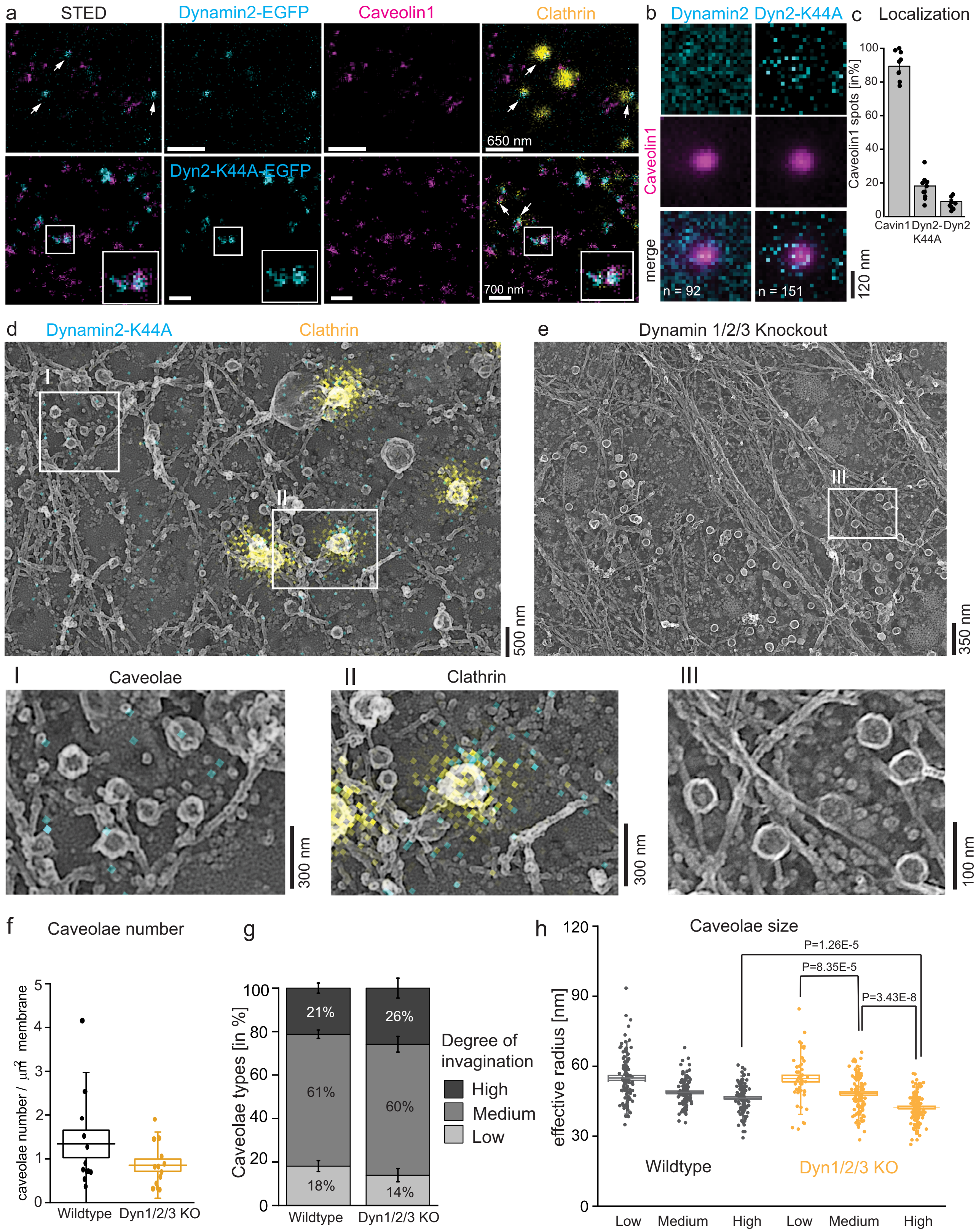


contrast imaging of Pt replicas was used to re-assign the imaged cells on the TEM grids. After TEM imaging the STED images were aligned to the TEM montages by using a Matlab code previously described ([46,85]). Brief alignment was obtained by arranging the cell borders visible in both images. Exact alignment of STED and TEM images was achieved by clathrin fluorescence staining position to their distinctive clathrin structures in the TEM image, as well as caveolin1 staining and caveolae structures in the replicas. Inspection and analysis of CLEM images was done in ImageJ/Fiji and Matlab accordingly to Sochacki et al. (2017)[46].

### STORM-CLEM

HeLa (Atcc Cat# CCL-2) STORM-CLEM data were all previously published in Sochacki et al., 2017[46] with the exception of Dyn2-K44A

**Fig. 8 | Dynamin is not localized to caveolae. a** Representative STED images of MEF membrane sheets expressing dynamin2-EGFP or dynamin2-K44A (in cyan), together with caveolin1 (magenta) and clathrin (yellow) immuno-staining. White arrows indicate dynamin localization. **b** Quantitative analysis of dynamin2 or dynamin2-K44A localization to caveolin1 spots in STED images illustrated in average STED fluorescence intensity projections (n(dynamin2) = 92, n(Dyn2-K44A) = 151, 3 independent experiments). **c** Percentage of caveolin1 spots that were also stained for cavin1, Dyn2-K44A or dynamin2. Bar plot indicates mean ± SE (n(cavin1) = 1135/8 cells, n(Dyn2-K44A) = 1283/11 cells, n(dynamin2) = 1032/8 cells, 3 independent experiments). **d** Representative STED-CLEM image showing Dyn2-K44A (cyan) and clathrin (yellow) on Pt replica TEM image. Increased image (**I**) illustrates caveolae, (**II**) shows clathrin vesicle (2 independent experiments). **e** Representative PREM image of membrane sheet obtained from dynamin triple

knockout (dynamin1/2/3) MEFs. Zoom (III) illustrates increased membrane area covered with caveolae. **f** Total caveolae number at the plasma membrane in wild-type and dynamin triple knockout MEFs. Box plot shows mean ± SE, whiskers illustrate SD (n(wt) = 39 cell regions, n(Dyn 1/2/3 KO) = 13 cell regions, 2 independent experiments). **g** Percentage of caveolae types observed in plasma membrane sheets of wild-type and dynamin triple knockout MEFs. Bar graph indicates mean ± SE (n(wt) = 39 cell regions, n(Dyn 1/2/3 KO) = 13 cell regions, 2 independent experiments). **h** Radius of individual caveolae types (round caveolae domain was assumed), box plot shows mean ± SE, whiskers illustrate SD (caveolae number: wt: n(low) = 100, n(medium) = 113, n(high) = 96; Dyn 1/2/3 KO: n(low) = 50, n(medium) = 110, n(high) = 121, 2 independent experiments, significant difference was tested by two-sided Mann–Whitney test).

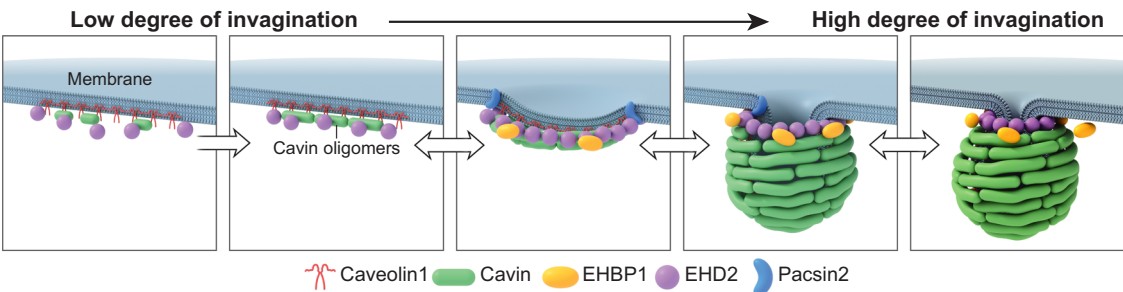

**Fig. 9 | Caveolae formation and bending at the plasma membrane.** Schematic model of caveolae formation and bending of the caveolae core complex.

expression in SK-MEL-2 cells (Atcc Cat# HTB-68). These data were acquired in the same manner as the previously published data using a Dyn2 (K44A)-GFP plasmid and staining with Alexa Fluor 647-conjugated GFP nanotrap[89]. The K44A point mutation was obtained with the Quickchange mutagenesis kit (Agilent #200523) and the following primer set: FWD- TGG GCG GCC AGA GCG CCG GCG CGA GTT CGG TGC TCG AGA; REV- TCT CGA GCA CCG AAC TCG CGC CGG CGC TCT GGC CGC CCA. Sequence was confirmed following mutation.

**Protein isolation and Western Blotting**

For protein isolation and Western blotting 100.000–200.000 cells were plated in 6 well plate and incubated for 48 h. After cells were washed with ice-cold PBS, 100 μl ice-cold RIPA buffer supplemented with proteases inhibitors (abcam #ab156034, Thermo Scientific #87786) was added for cell lysis, and lysed cells were transferred in 1.5 ml tubes for vortexing. Next, cell lysates were incubated for 30 min on ice followed by 10 min centrifugation (10,000 × g) at 4 C. 10 μl of the supernatant was used for SDS-PAGE on 4–12% Tris-Glycine gels (NOVEX™, Invitrogen #XP04120BOX) with Tris-Glycine SDS Running buffer (NOVEX™, Invitrogen #LC2675) at 120 V. Western Blotting was performed by MiniBlot (iBLOT, Invitrogen #IB1001) with ready-to-use membranes (NOVEX™, Invitrogen #IB401002) accordingly to the manufacturers protocol. Afterwards, membranes were incubated for 1 h at room temperature in 5% milk/TBS-T (1% Tween-20 diluted in TBS, NOVEX™ #28358). Primary antibodies were diluted in 5% milk/TBS-T and applied on the membranes over night at 4 C (on horizontal shaker). After 3 times 10 min washing with TBS-T secondary antibody solution was added for 2 h at room temperature. ECL solution (Amersham, GE Healthcare #RPN2232) was used for detection of protein levels in Chemi-Doc XRS system (Biorad #1708265).

Antibodies: anti-Dynamin 1/2/3-mouse (BD Science #610245, 1:1000), anti-Pacsin2-Rabbit (Proteintech #10518-2-AP, 1:500), anti-EHBP1-Rabbit (Proteintech #17637-1-AP, 1:1000), anti-GAPDH-Rabbit (Cell Signaling #8884, 1:1000), anti-EHD2-Rabbit (abcam #ab222888, 1:500), anti-mouse-HRP (dianova #115-035-146, 1:5000), anti-Rabbit-HRP (dianova #111-035-045, 1:5000).

**Statistical analysis**

All statistical analysis was carried out in Origin 2018b. First, data sets were analyzed by descriptive statistics (mean, standard error of the mean (SE), median, min, max, standard derivation (SD), 5–95% interval) and normal distribution was tested by Shapiro–Wilk and Kolmogorov–Smirnov test. If data sets were normally distributed statistical differences were evaluated by two-tailed $t$-test, otherwise Mann–Whitney test was applied (significance level 0.05, exact $P$ value was measured). The following range of statistical differences is used in all figures: * $P < 0.05$, ** $P < 0.01$, *** $P < 0.001$, **** $P < 0.0001$.

## Data availability

The data generated in this study has been deposited in Figshare at https://doi.org/10.25444/nhlbi.c.6253644. The remaining data are available in the paper or Supplementary Information files. Source data are provided with this paper.

## Code availability

MATLAB codes used in this study are specific to lab file formatting. The codes are available in Figshare at https://doi.org/10.25444/nhlbi.14502156.v1.

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

## Acknowledgements

We thank the NHLBI electron microscopy and light microscopy cores, and the FMP light and electron facility for their support and technical assistance. We thank Dr. Pietro De Camilli for the kind gift of the dynamin triple knockout MEFs. We thank Fabian Lukas and Tania López-Hernández for providing C2C12 myoblast cells and mouse astrocytes. We thank the Taraska lab for critical evaluation of experiments and paper. J.W.T. is supported by the Intramural Research Program of the National Heart Lung and Blood Institute, National Institutes of Health. V.H. acknowledges funding by the Deutsche Forschungsgemeinschaft (CRC958/A01).

## Author contributions

C.M., M.L., and J.W.T. designed and discussed the experiments. C.M. performed and analyzed all experiments. K.A.S. wrote MATLAB code for analysis of CLEM and STED data, supported and discussed analysis, performed STORM-CLEM and PREM of HeLa. A.M.D. did STORM-CLEM of Sk-mel2 cells. D.P. supported EM. V.H. supported and discussed experiments. C.M. and J.W.T. wrote the paper with input from all authors.

## Competing interests

The authors declare no competing interests.
