## [Peer Review File · Nature Communications]

REVIEWER COMMENTS

Reviewer #1 (Remarks to the Author):

This study uses correlative EM combined with STED (STED-CLEM) to define how the main caveolar components distribute/localize in the different curvatures of the structure and in the flattened version. This study provides an alternative picture to the current view in the field, especially regarding the role of cavins with respect to the curvature of caveolae. The authors find that cavins are still present in flattened caveolae, contrary to the current view in the field. In addition, the involvement of dynamin2 in caveolae endocytosis is also challenged. Details about EHBP1, EHD2 and pacsin2 recruitment to different caveolar stages are also provided.

This is an interesting and relatively well conducted study that reveals important aspects of caveolae biology. However, substantial additional experimentation is needed to sustain their claims. The study is based on a single technique, to provide alternative views regarding the biology of caveolar regulators (mostly cavins and dynamin) and this is a limitation, although the technique seems to be quite powerful. In order to challenge the relatively extensive literature on those caveolar regulators, several issues must be fully addressed, so that there is no doubt that these observations are truly meaningful and well interpreted.

- 1) Regarding the classification of caveolae. It is unclear how classification is made on a caveolae-caveolae basis. Was each structure classified based on the intensity plot or based on visual subjective observation? How were the structures selected to create figure 2D? Does any caveolae intensity profile fall within any of the differentiated intensity profiles depicted in figure 2d? Was the radius used to classify objects? The classification method should be clearly explained and defined. This is important, as in some cases (compare fig 2b, MEF panel flat vs bulb, or Fig 4A, bulb vs sphere, or in the same panel, flat vs bulb), the visual inspection does not allow to differentiate both structures. Related to the classification, it is unclear which criteria is used to define a flattened caveolae. In figure 4 a flattened caveolae is labelled with cavin1 (D), cavin2 (E) and cavin3 (F), but those structures appear to be bulb caveolae, or at least curved structures. Cavin1-3 signal is also observed in assemblies, in this case it is more obvious that is not curved. It is essential to fully clarify these issues and define in a precise manner the boundaries between the different structures. Terminology should be clearly defined/unified throughout the figures and text. "assembly", "flat assembly", "flat" and "immature caveolae" terms are used and is not clear the difference between these structures.
- 2) The effect of hypo-osmotic shock on the abundance of bulb and flat caveolae shows that the effect of osmotic swelling is moderate, while other studies have shown stronger effects. This could be due to osmolarity differences (here 1/5 dilution, while other studies use 1/10 dilution). The authors should use 1/10 dilution and include in their quantification all the pools (assembly -or whatever terminology is finally used-, flat, bulb and sphere), to have a precise picture of how these caveolae phases behave upon increased tension. It is confusing why in some cases all categories are analyzed while in other cases only some are analyzed.
- 3) A striking finding of this study is the presence of Cavins in flattened caveolae, which is not consistent with other studies and the current dogma. It is important to use the same conditions (1/10 dilution for hypo condition), label endogenous Cavin1 in iso and hypo and quantify the cavin1 labeling in the different caveolae stages/phases of curvature -assembly, flat, bulb and sphere-. This quantification could be then replicated with Cav1 staining and determine more precisely whether Cavin1 is released when caveolae flatten out. Since this is the strongest piece of data of the manuscript, without a clear understanding of the reasons that explain the contradictory views in the field it would be difficult to fully appreciate these interesting observations.
- 4) Line 72. "it is unclear ...if caveolae disassemble upon increased tension". This statement should be placed in context of other studies suggesting the opposite.
- 5) Line 75: Could the authors specify what they mean by "these sites". It is now well established that pacsin2 and Ehd2 are recruited to caveolar neck. At the end of the sentence it is stated "and its shape": again, the shape of what, the caveolin/cavin coat? Some sentences are a bit vague and omit other studies.
- 6) Why endogenous cavin1 is not labelled in fig 3c?
- 7) The staining of endogenous EHBP1 is a bit dispersed and out of caveolae, this would require confirmation with additional antibodies or validation of the specificity of the current antibody.
- 8) This study focuses on the different caveolae configurations but rosettes are not studied. These

clusters are highly abundant in adipocytes, endothelial cells and relatively frequent in other cell types. Were they omitted in the analysis or were not detected/present? Can they be detected with this technique? This should be clarified and discussed. In case this technique does not allow to detect them (it is surprising that are not observed in the figures) it is a limitation for the interpretation of the results and should be considered.

9) In figure 6 only one siRNA is used, not rescue experiments nor additional siRNA sequences, this should be fixed.

10) The authors claim that dynamin is not recruited to caveolae nor regulates the abundance of caveolae. Related to a comment above, what if dynamin is involved in rosettes or recruited to them? Or only in induced -and not basal- endocytosis of caveolae? If the authors want to discard dynamin as a regulator of caveolae biology once and for all, additional experimentation is needed to fully sustain their claims. As the authors state, "future work is needed".

11) The authors argue that spheres may represent the last step before endocytosis, this means that in their cells there could be significant endocytosis (as about a third of caveolae are in that stage/configuration), yet endocytosis of caveolae in basal conditions is limited. This should be discussed.

12) The final paragraph of the discussion is repetitive. In general, the discussion would benefit from some editing.

13) It is unclear whether all experiments were carried out at least 3 independent times. This should be confirmed in each figure legend, now it is not indicated in many figures.

Minor points:

-Western blots in fig 6a. Images are a bit messy. GAPDH is not clearly seen in the last blot. It seems that pacsin2 siRNA affects the levels of EHBP1, is this an off-target effect? Should be quantified and discussed if true.

-Where is fig S6C?

-Fig 1a and related text. The terms MEF and fibroblast are used for the same cell type, please unify terms.

-Line 213-214-215 : references are missing.

Line 214: "and increase" should read an increase.

-Fig 5d. the magenta caveolin legend is hard to read (at least to my eyes).

- Fig S4c, graph says bulb, fig legend says bulb+sphere. Please unify terminology.

Reviewer #2 (Remarks to the Author):

This manuscript describes the systematic analysis of caveola morphology and protein localisation using a powerful combination of platinum replica electron microscopy (PREM) and correlated stimulated emission depletion (STED) fluorescence microscopy. The approach uses unroofed plasma membrane sheets of various cell lines and compares the localisation of core caveola proteins CAV1, CAV2 cavin1, cavin2, cavin3 with regulatory proteins EHD2, EHBP1, pacsin and dynamin and the impact of inhibiting these regulatory proteins on caveola formation. The work provides a high-resolution validation of previous studies of caveola protein organisation, but also presents several novel insights that will be of interest to the field. These novel insights include (i) that dynamin does not appear to play an active role in caveola assembly or membrane scission, (ii) that when caveolae are flattened (either naturally, or by inducing membrane stretch) CAV1 and cavin1 remain closely associated, (iii) the EHD2 protein is localised to essentially all caveola structures including flat caveolae and not only to the neck of assembled caveola vesicles, (iv) that EHBP1 protein and pacsin are recruited to only a subset of caveolae and with distinct curvature preferences.

I thought the paper was well written, with appropriate limitations outlined, the data was generally very well presented and highly convincing, and overall the work provides interesting new insights into the structural organisation of caveolae. For context, I have expertise in the structural and cell biology of membrane-associated proteins, but not specific expertise in the EM or STED imaging methods described. I have only a few relatively minor queries and edits to suggest.

Queries.

1. Pg. 5 and Fig. 3. It is mentioned that cavin1 and EHD2 are localised to the majority of CAV1-positive caveolae while pacsin and EHBP1 are more restricted. What about cavin2 and cavin3? Although their distribution within individual caveolae is similar to cavin1, are they found in all caveolae or just a subset?
2. Related to the first query, although the data supports the model that cavin1 remains associated with CAV1 even when caveolae are flattened, have you looked at either cavin2 or cavin3? Cavin3 in particular has been proposed to be specifically relocalised to the cytoplasm and nucleus under cell stress to interact with other proteins such as BRCA1 (McMahon et al., 2021, eLife).
3. On the same lines, have the authors ever examined the proportion of flat to curved structures, or the co-localisation of CAV1 and cavin1 with other caveola proteins following cholesterol depletion? I am not suggesting this experiment needs to be done for this paper, but if it has been done the results would be worth including or at least commenting on.
4. Figure S6 shows blots of the pacsin and EHBP1 proteins in different knockdown and knockout cells. However EHD2 is not included in any of these blots. I realise the EHD2 KO line was generated in a previous study, but it would be good to show in these particular cells that EHD2 is no longer expressed. Also there does not appear to be a Fig. S6C showing CAV1 and cavin1 levels in the version I have reviewed.
5. This may merely be my own biased view of the images, but it often appears that caveolae are associated with filamentous protein densities. Are these microtubules? And is there any sense that caveolae (perhaps in contrast to clathrin-coated vesicles) are mostly found in association with these structures?
6. In Figure 1, when quantifying the number of stripes and the length of stripes, it would be good to see a zoomed example (perhaps as supplementary) of one of the measured stripes to provide a visual representation as to how the cut-offs between the beginning and end of the stripes are determined. I realise this will be somewhat subjective, but it would be useful to see an example. Also would it be possible to provide a measurement of either the stripe widths, or at least the spacing between adjacent stripe structures? Perhaps by simply drawing a line through a few representative caveolae and plotting the intensity versus distance and calculating an average distance between peaks?

Minor points:

7. Pg. 4, line 133. 'Morphologically' should be 'morphological'.
8. In Fig. 2A what is the difference between a 'flat assembly' and 'flat' caveolae?
9. Pg. 6, line 214. 'in and increase' should be 'in an increase'.
10. Pg. 8, line 305. 'with an mild' should be 'with a mild'.

Reviewer #3 (Remarks to the Author):

This manuscript aims to describe the formation of caveolae and assign a molecular signature to each of the 3 stages identified. They mainly use a STED-PREM correlative workflow to gather their data. The authors have acquired an impressive amount of data using this advanced CLEM method to be able to generate quantitative data.

Comments:

- The very first section of the Introduction is very much based on reviews rather than original publications. e.g. the generation of cave-in-1 KO mice.
- Page 3, line 80 refers to EM studies but no references are mentioned.
- Page 4, line 147. I was surprised to see that the spheres are actually not grape-shaped as in traditional TEM images. Is this a consistent theme or just in this example. It may be good to indicate arrows the overview tomogram to highlight the zoomed areas.
- Page 4, line 154 sounds a bit like this is a new finding in this paper which is well known. Might want to rephrase.
- Page 5, Line 182: associated
- Page 5, Line 187: What about the unlabelled "caveolar-like structures" in first zoom just above and below Cathrin labelled structures. The fluorescence for cavin in zoom "a" actually looks shifted.
- Page 5, Line 188, Fig 4A,B: These images contradict the statement that the size of caveolae in STED is the same as PREM. STED fluorescence is clearly wider distributed. Not just at the formation stage.
- Page 6, Line 209: First images of figures 4, 5, 6B do not show correct scale bar. Would also have

been "nice" to have a consistent way of depicting those.

- Page 7, Line 231: Although this is briefly discussed later: As caveolae are dynamic how can a distinction be made between forming and collapsing / merging structures. Is the protein composition the same for both. As mentioned by the authors, mild osmotic shock and addition of lipids could discriminate between the 2. This is something that is not really addressed by the authors.

- Page 7, Line 260: I disagree with the conclusion of the authors that there is no Dynamic colocalization with caveolae. I see a clear colocalization of the Dynamic K44A mutant in the Zoom sections. The argument that those would be Cathrin structures because of the few yellow spots would be a very selective interpretation.

This last point is my main issue with this manuscript.

Otherwise the manuscript is well written, has a logical flow with very impressive quantitative imaging data.

We thank all three reviewers for their helpful comments and suggestions to improve the manuscript. Based on the comments, we have made the following key changes:

- 1) Revised our terminology regarding the classification of caveolae curvature and added a more detailed description—along with additional analysis—of how caveolae were segmented and grouped into different classes (new fig. 2, suppl. fig. S3).
- 2) Performed and added new STED-CLEM and two-color TIRF experiments to analyze cavin localization to caveolae before and after both strong and mild osmotic shock (new fig. 4, suppl. fig. S7-9).
- 3) Done and included additional dynamin localization experiments (suppl. fig S12).
- 4) Included new representative images of caveolae rosettes (new suppl. fig. 1).
- 5) Added a more detailed presentation of caveolae coat measurements (new suppl. fig. S2).
- 6) Performed and added new experiments and analysis for the localization of cavin2 and 3 at individual caveolae sub-types (fig. 4 and 5).

Below, please find a point-by-point response (black text) to all specific comments (blue text). Thank you for your time and efforts in this review. We hope the paper is now acceptable for publication.

REVIEWER COMMENTS

Reviewer #1 (Remarks to the Author):

This study uses correlative EM combined with STED (STED-CLEM) to define how the main caveolar components distribute/localize in the different curvatures of the structure and in the flattened version. This study provides an alternative picture to the current view in the field, especially regarding the role of cavin with respect to the curvature of caveolae. The authors find that cavins are still present in flattened caveolae, contrary to the current view in the field. In addition, the involvement of dynamin2 in caveolae endocytosis is also challenged. Details about EHBP1, EHD2 and pacsin2 recruitment to different caveolar stages are also provided.

This is an interesting and relatively well conducted study that reveals important aspects of caveolae biology. However, substantial additional experimentation is needed to sustain their claims. The study is based on a single technique, to provide alternative views regarding the biology of caveolar regulators (mostly cavins and dynamin) and this is a limitation, although the technique seems to be quite powerful. In order to challenge the relatively extensive literature on those caveolar regulators, several issues must be fully addressed, so that there is no doubt that these observations are truly meaningful and well interpreted.

We thank the reviewer for their positive and helpful review of our manuscript and are thankful for the comments and suggestions.

1) Regarding the classification of caveolae. It is unclear how classification is made on a caveolae-caveolae basis. Was each structure classified based on the intensity plot or based on visual subjective observation? How were the structures selected to create figure 2D?

The classification of caveolae were made by visual inspection of individual caveolae coat curvatures and the relative intensity of the caveolae edge signal to the surrounding membrane. Along with the intrinsic curvature in the coat, when comparing caveolae with a lower degree of invagination to those with medium or highly curved coats, the edge of the organelle appears strongly white in the TEM images when the organelle is highly curved. This signal is generated during rotary platinum coating when metal accumulates along the side-wall of these relatively tall caveolae. This additional metal blocks passing electrons and results in a white signal in inverted TEM images. This signal is missing in organelles with low curvature. This type of analysis and signal has been used by our lab in several past papers (Sochacki et al, Nat. Meth 2014, Nat. Cell Bio 2017, Dev Cell 2021, Prasai et al. Nat. Comm 2021, Alfonso Mendez et al. Nat. Comm 2021). To confirm the general applicability of our manual classifications, we have now quantitated the grey intensity value difference between the edge and surrounding membrane and compared these values to the average manual classification subgroups (new suppl. fig. S3, line 157-165). When low, medium, and highly curved caveolae were segmented and grouped in MEFs, HUVEC or HeLa cell replicas, the low curved caveolae group shows reduced intensity difference between the edge and its surrounding membrane. With increasing degrees of invagination, the intensity difference increases, indicating that the visual inspection of caveolae and its average classification mirrors our average intensity measurements. Importantly, this type of classification works best with large numbers of structures and cells, allowing us to accommodate the small number of possible miss-assignment. Yet, to fully explore these potential errors, we now present new segmented replica membranes that were also imaged in 3D electron tomograms where we can directly measure the axial height of the same caveolae (new fig. 2c). As expected, caveolae manually segmented with low curvature had the lowest measured heights, the other two groups showed taller caveolae. Heights and grey value differences of individual segmented caveolae strongly correlated (new suppl. fig. 3f) further supporting the idea that visual classification is suitable to distinguish caveolae curvature types. We acknowledge that for future work, it would be useful and faster to implement a fully automatic segmentation tool (possibly AI-based) to classify different caveolae types without a user's input. However, AI tools would still rely on human-based training data generated by manual segmentation of curvature subtypes and would contain similar biases. Likewise, non-AI based intensity measurements are very noisy on a single vesicle level and are greatly impacted by the complex signals generated from nearby structures in the crowded cellular environment. This makes it very difficult to develop a robust automatic quantitative metric for grouping that is better or more reliable than human-based visual inspection. Thus, currently no automatic classification of caveolae curvature is available. We feel that the development of this analysis is beyond the scope of this work. We also feel that visual classification is a straight-forward method to inspect large plasma membrane areas and generate average groupings for caveolae curvature sub-types. This analysis is now more fully described and presented in the text and methods.

Does any caveolae intensity profile fall within any of the differentiated intensity profiles depicted in figure 2d? Was the radius used to classified objects?

Yes, individual caveolae can reside between the different intensity profiles depicted in new suppl. fig. 3b. We did not use radius in our classification. The radius was measured (and shown to be smaller) after segmentation (fig. 2e). Additional factors are used when manually classifying caveolae which we now describe in more detail in the text.

The classification method should be clearly explained and defined. This is important, as in some cases (compare fig 2b, MEF panel flat vs bulb, or Fig 4A, bulb vs sphere, or in the same panel, flat vs bulb), the visual inspection does not allow to differentiate both structures. Related to the classification, it is unclear which criteria is used to define a flattened caveolae. In figure 4 a flattened caveolae is labelled with cavin1 (D), cavin2 (E) and cavin3 (F), but those structures appear to be bulb caveolae, or at least curved structures. Cavin1-3 signal is also observed in assemblies, in this case it is more obvious that is not curved. It is essential to fully clarify these issues and define in a precise manner the boundaries between the different structures. Terminology should be clearly defined/unified throughout the figures and text. “assembly”, “flat assembly”, “flat” and “immature caveolae” terms are used and is not clear the difference between these structures.

Thank you for this important comment. We agree that the original terminology was not clear. In the revised manuscript, we now use the more general terminology of “low, medium, or high” curved caveolae (new fig. 2, suppl. fig S3). Indeed, even flat caveolae at this scale have some degree of intrinsic thickness and slight curvature. The metal coating adds to this ambiguity. However, compared to medium or highly curved caveolae, the low curved caveolae show a greatly reduced height as seen in the electron tomogram fig. 2 c. Likewise, the term bulb is not clear. We no longer use this term. We have also removed the “assembly” state from the manuscript because we cannot ascertain if these events are newly formed/assembling caveolae or proteins that remain after disassembly or caveolae endocytosis. Furthermore, these structures are quite heterogenous which makes them unsuited for quantitative analysis. Now, we show them in the supplement for interested readers (new suppl. fig S6). We hope these terms are more clear.

2) The effect of hypo-osmotic shock on the abundance of bulb and flat caveolae shows that the effect of osmotic swelling is moderate, while other studies have shown stronger effects. This could be due to osmolarity differences (here 1/5 dilution, while other studies use 1/10 dilution). The authors should use 1/10 dilution and include in their quantification all the pools (assembly - or whatever terminology is finally used-, flat, bulb and sphere), to have a precise picture of how these caveolae phases behave upon increased tension. It is confusing why in some cases all categories are analyzed while in other cases only some are analyzed.

We thank the reviewer for this comment and agree that it would be interesting to use osmotic shock to compare our work directly to past work done with extremely strong osmotic swelling. Thus, we now include a new experiment with extreme osmotic shock (10 fold dilution) as previously applied by Sinha et al., Cell 2011; McMahon et al., elife 2021 or Torino et al., JCB 2018. As shown in the new fig. 5c with EM, the total caveolae number was slightly reduced after HUVEC cells were treated with strong osmotic shock, which was not observed under mild osmotic shock (suppl. fig. S7c). Slightly more low curved caveolae were observed after strong osmotic shock (Fig. 5d) similar to mild osmotic shock (suppl. fig S7c). The proportion of medium or highly curved caveolae, however, were not changed. This is contrary to previously published work (Sinha et al). It should be noted that past work did not quantitate caveolae curvature

directly. Rather, the authors measured caveolae protein localization by fluorescence. Sinha et al. (2011) showed one example Pt replica image after strong osmotic shock treatment. This image contained some low curved caveolae (Sinha et al., Cell 2011, fig. 3D, sup fig. S2). A detailed inspection of this TEM image, however, also shows medium and highly curved caveolae in the membrane region. Furthermore, the authors did not quantitate the distribution or ratios of low and highly curved caveolae in platinum replicas across many cell membranes as we have now presented.

<https://www.sciencedirect.com/science/article/pii/S0092867410015230?via%3Dihub#figs2>

Lastly, the use of the STED-CLEM allows us to detect all caveolae in Pt replicas after osmotic shock which again supports our observations.

3) A striking finding of this study is the presence of Cavins in flattened caveolae, which is not consistent with other studies and the current dogma. It is important to use the same conditions (1/10 dilution for hypo condition), label endogenous Cavin1 in iso and hypo and quantify the cavin1 labeling in the different caveolae stages/phases of curvature -assembly, flat, bulb and sphere-.

We now included STED-CLEM of endogenously-labeled Cavin1, 2 and 3 (antibody labeling) in plasma membrane sheets of HUVEC before and after both strong osmotic shock (10 fold dilution, 1:9, new fig. 5 and suppl. fig. S8) and mild osmotic shock (6 fold dilution, 1:5, new suppl. fig. 7; lines 221-234). Similar to our data with expressed proteins, we detect endogenous cavin proteins before and after strong or mild osmotic shock at low curved caveolae. Surprisingly, we observe that highly curved caveolae tend to contain slightly less cavin. However, STED microscopy cannot evaluate exact protein numbers due to bleaching and other factors. Thus, our findings do not support the model that flat caveolae do not contain cavins as suggested in past models. This is now discussed in detail in the text (line 350-352).

This quantification could be then replicated with Cav1 staining and determine more precisely whether Cavin1 is released when caveolae flatten out. Since this is the strongest piece of data of the manuscript, without a clear understanding of the reasons that explain the contradictory views in the field it would be difficult to fully appreciate these interesting observations.

Thank you for this comment. To further verify our STED-CLEM data, we now include TIRF microscopy of intact HUVEC cells before and after strong or mild osmotic shock (new suppl. fig. S8). We stained endogenous Cavin1, 2 and 3 together with caveolin1. Similar to our previously-reported data, we detected all three cavin proteins colocalizing with caveolin1 spots. We did not observe a substantial reduction in the number of caveolae at optical resolution in cells after osmotic shock. Furthermore, we measured fluorescence intensity for cavin1-3 and caveolin1 and did not detect disassembly or a measurable reduction of these proteins in the population after osmotic shock with two-color TIRF. We could not classify these signals according to curvature because we used intact cells and could not do EM on these samples.

4) Line 72. "it is unclear ...if caveolae disassemble upon increased tension". This statement should be placed in context of other studies suggesting the opposite.

Thank you for this comment, we rephrased this sentence (line 72).

5) Line 75: Could the authors specify what they mean by “these sites”. It is now well established that pacsin2 and Ehd2 are recruited to caveolar neck. At the end of the sentence it is stated “and its shape”: again, the shape of what, the caveolin/cavin coat? Some sentences are a bit vague and omit other studies.

We agree that this sentence was vague. We rephrased the sentence and now use a different curvature description throughout the manuscript.

6) Why endogenous cavin1 is not labelled in fig 3c?

We used egfp-tagged caveolae proteins in the STED experiments to better match the size of the biological objects. In the revised manuscript, we have added new antibody-labeled cavin1 and caveolin1 STED data to mark endogenous proteins (new suppl. Fig. S4e). As expected, both proteins colocalize as shown in the confocal and STED images.

7) The staining of endogenous EHBP1 is a bit disperse and out of caveolae, this would require confirmation with additional antibodies or validation of the specificity of the current antibody.

The antibody used to detect endogenous EHBP1 was KO/KD validated by the supplier:

<https://www.ptglab.com/products/EHBP1-Antibody-17637-1-AP.htm>

We used the same antibody to evaluate EHBP1 siRNA knockdown efficiency (suppl. fig. S11b). We conclude that the antibody is suitable for detecting EHBP1 at the plasma membrane. When comparing immunostaining endogenous protein with EHBP1-EGFP expressing cells, we observed similar staining profiles at the plasma membrane, further validating this particular EHBP1 antibody.

8) This study focuses on the different caveolae configurations but rosettes are not studied. These clusters are highly abundant in adipocytes, endothelial cells and relatively frequent in other cell types. Were they omitted in the analysis or were not detected/present? Can they be detected with this technique? This should be clarified and discussed. In case this technique does not allow to detect them (it is surprising that are not observed in the figures) it is a limitation for the interpretation of the results and should be considered.

Yes, caveolae rosettes can be observed in platinum replicas. We now include representative TEM images of rosettes found in membrane sheets in MEFs, HeLa, HUVEC and adipocytes (new suppl. fig. S1, line 118-119). We excluded caveolar rosettes from our analysis because their size and composition varied and therefore made it difficult to extract average structural information. We hope the inclusion of these images will be useful to those who are interested in caveolae rosettes.

9) In figure 6 only one siRNA is used, not rescue experiments nor additional siRNA sequences, this should be fixed.

Thank you. We did not use a single siRNA but instead used a pool of 4 siRNAs (Dharmacon SMARTpool). We apologize if this was not clear. We now highlight this in the text (line 267, 269). We did not rescue the siRNA knockdown. Rescue experiments would require extensive

reworking of our expression plasmids and multi-part validation experiments which are outside the core findings of the paper. However, the effect of pacsin loss on caveolae is well described in previous articles from the Qualmann lab. We observed similar changes in caveolae as shown in their pacsin3 KO (muscle-specific KO mouse, Seemann et al., 2017) or pacsin2 KD cells (Koch et al., 2012). EHD2 loss was described by us and other groups previously in cell culture and in vivo (Moren et al., 2012, Stoeber et al., 2012, Matthaeus et al, 2022). Re-expression of EHD2 reduces caveolae mobility and stabilizes caveolae at the plasma membrane. In the current manuscript, we include EHD2-EGFP overexpressing STED-CLEM data (fig. 6) in comparison to endogenous EHD2 antibody staining (suppl. fig. SS8). As we did not detect differences, we omitted a rescue experiment for EHD2 KO cells. EHBP1 knockdown did not have any measurable effect and thus again we did not perform a rescue experiment.

10) The authors claim that dynamin is not recruited to caveolae nor regulates the abundance of caveolae. Related to a comment above, what if dynamin is involved in rosettes or recruited to them?

Thank you for this thoughtful comment. We re-analyzed our CLEM data for potential dynamin localization to caveolar rosettes. In HeLa cells dynamin2 and MEF Dynamin2-K44A mutant cells we could not detect strong localization to rosettes. We conclude that at caveolar rosettes, dynamin does not accumulate. See below for example images (orange arrows indicate rosettes and magenta is dynamin).

HeLa
Dynamin2-GFP
(STORM-CLEM)

100 nm

100 nm

270 nm

MEF
Dynamin2-K44A-GFP
Clathrin
(STED-CLEM)

200 nm

150 nm

Or only in induced -and not basal- endocytosis of caveolae?

We now include a new set of STED experiments where we localize dynamin2 in MEFs treated with oleic acid for 30 or 60 min (new suppl. fig. S12d). Lipids are known to induce endocytosis of caveolae (e.g.: Hubert et al., *elife* 2020; Matthaeus et al., *PNAS* 2020). Pearson correlation analysis of caveolin1 (as marker for caveolae) and dynamin did not reveal an increase after oleic acid treatment (suppl. fig S12d). Therefore we conclude that increased caveolar mobility and endocytosis caused by external lipid treatment fails to measurably recruit dynamin to caveolae. However, we cannot exclude the possibility that dynamin may be recruited during caveolae endocytosis triggered by other factors. Future work is needed.

If the authors want to discard dynamin as a regulator of caveolae biology once and for all, additional experimentation is needed to fully sustain their claims. As the authors state, “future work is needed”.

We agree that much work is needed to determine the functional role of dynamin in caveolae. However, based on our data, we propose that dynamin may not act directly on caveolae, but rather on other factors such as actin. This is now presented in the discussion. Notably, when we viewed dynamin triple KO MEFs, caveolae number and structure were not altered. It is reasonable to assume that if dynamin plays a direct role in caveolae membrane scission, we would have detected some changes (Fig. 8e-f). Likewise, in preliminary TIRF experiments, we observed that caveolae mobility and endocytosis was not impaired in MEFs lacking dynamin (data not shown). However, recently Larsson et al. (*BioRxiv* 2022) reported that dynamin loss increased caveolae mobility which may suggest that other mechanism play a role in this process in some cells. Based on our CLEM and STED data from multiple cell types, however, we propose that dynamin does not (or very rarely, >5% of all caveolae) localize to caveolar membrane domains. Yet, we cannot exclude the idea that specific cellular triggers could induce dynamin accumulation or this association is very transient or very sparse. This is now discussed in the text.

11) The authors argue that spheres may represent the last step before endocytosis, this means that in their cells there could be significant endocytosis (as about a third of caveolae are in that stage/configuration), yet endocytosis of caveolae in basal conditions is limited. This should be discussed.

Thank you for this comment. We agree that we cannot evaluate if highly curved caveolae undergo membrane scission or instead flatten or disassemble. In these instances, 3D temporal data (such as TIRF in live cells) correlated to PREM is needed. This may explain the discrepancy to previous work in live-cells. We now discussed this in the text (line 327-331). Live cell CLEM is currently unfeasible with our experimental pipeline.

12) The final paragraph of the discussion is repetitive. In general, the discussion would benefit from some editing.

We removed the final paragraph from the discussion and have re-structured the text. We hope the revised manuscript is now more accessible.

13) It is unclear whether all experiments were carried out at least 3 independent times. This should be confirmed in each figure legend, now it is not indicated in many figures.

We apologize for missing this information. We now include the number of independent experiments for the treatments or experiments in all the figure legends.

Minor points:

-Western blots in fig 6a. Images are a bit messy. GAPDH is not clearly seen in the last blot. It seems that pacsin2 siRNA affects the levels of EHBP1, is this an off-target effect? Should be quantified and discussed if true.

We improved the western blots and present them based on the individual proteins EHD2, pacsin2, or EHBP1 (new suppl. fig. S11). We also repeated the EHBP1 western blot to detect any differences in EHBP1 protein level in pacsin2 KD MEFs. However, no strong changes for EHBP1 were observed. EHD2 KO cells showed a slight reduction in EHBP1, however, as we did not observe any effect on caveolae curvature and number in EHBP1 KD cells, we conclude that this should impact the reported results for EHD2 KO (or pacsin2 KD) cells illustrated in fig. 7.

-Where is fig S6C?

We apologize for this mistake and have included the missing caveolin1 and cavin1 western blot (suppl. fig. 11d).

-Fig 1a and related text. The terms MEF and fibroblast are used for the same cell type, please unify terms.

Thank you. We changed the text accordingly.

-Line 213-214-215 : references are missing.

These have been added (now line 243).

Line 214: "and increase" should read an increase.

We changed this accordingly.

-Fig 5d. the magenta caveolin legend is hard to read (at least to my eyes).

We changed this.

- Fig S4c, graph says bulb, fig legend says bulb+sphere. Please unify terminology.

We apologize for this mistake and have changed our terms throughout the manuscript.

Reviewer #2 (Remarks to the Author):

This manuscript describes the systematic analysis of caveola morphology and protein localisation using a powerful combination of platinum replica electron microscopy (PREM) and correlated stimulated emission depletion (STED) fluorescence microscopy. The approach uses unroofed plasma membrane sheets of various cell lines and compares the localisation of core caveola proteins CAV1, CAV2 cavin1, cavin2, cavin3 with regulatory proteins EHD2, EHBP1, pacsin and dynamin and the impact of inhibiting these regulatory proteins on caveola formation. The work provides a high-resolution validation of previous studies of caveola protein organisation, but also presents several novel insights that will be of interest to the field. These novel insights include (i) that dynamin does not appear to play an active role in caveola assembly or membrane scission, (ii) that when caveolae are flattened (either naturally, or by inducing membrane stretch) CAV1 and cavin1 remain closely associated, (iii) the EHD2 protein is localised to essentially all caveola structures including flat caveolae and not only to the neck of assembled caveola vesicles, (iv) that EHBP1 protein and pacsin are recruited to only a subset of caveolae and with distinct curvature preferences.

I thought the paper was well written, with appropriate limitations outlined, the data was generally very well presented and highly convincing, and overall the work provides interesting new insights into the structural organisation of caveolae. For context, I have expertise in the structural and cell biology of membrane-associated proteins, but not specific expertise in the EM or STED imaging methods described. I have only a few relatively minor queries and edits to suggest.

We thank the reviewer for their helpful analysis of our manuscript and appreciate the comments that were raised. Please see below for detailed answers to all points raised.

Queries.

1. Pg. 5 and Fig. 3. It is mentioned that cavin1 and EHD2 are localised to the majority of CAV1-positive caveolae while pacsin and EHBP1 are more restricted. What about cavin2 and cavin3? Although their distribution within individual caveolae is similar to cavin1, are they found in all caveolae or just a subset?

Thank you very much for this comment. We now included analysis of cavin1, 2 and 3 regarding their localization to the different caveolae curvature types in MEF (fig. 4) and HUVEC (new fig. 5). We observed cavin2 and 3 at low, medium, and highly curved caveolae. However, we detected a slight trend where highly curved caveolae exhibited a modest reduction in cavin protein localizations. This trend was more pronounced in HUVEC (before or after osmotic shock, fig. 5) than in MEFs (fig. 4). As we used STED to visualize cavin localization, however, it is challenging to evaluate absolute cavin levels at single sites across many cells.

2. Related to the first query, although the data supports the model that cavin1 remains associated with CAV1 even when caveolae are flattened, have you looked at either cavin2 or cavin3? Cavin3 in particular has been proposed to be specifically relocalised to the cytoplasm and nucleus under cell stress to interact with other proteins such as BRCA1 (McMahon et al., 2021, eLife).

We now included STED-CLEM data from cavin2 and 3 after strong and mild osmotic shock in HUVEC (new fig. 5 and suppl. fig. 7). Cavin2 and 3 localize to low and highly curved caveolae independently of osmotic shock. However, as described above, we detect a slightly lower prevalence of cavin2 and 3 at highly curved caveolae (new fig. 5). This is discussed in the text.

3. On the same lines, have the authors ever examined the proportion of flat to curved structures, or the co-localisation of CAV1 and cavin1 with other caveola proteins following cholesterol depletion? I am not suggesting this experiment needs to be done for this paper, but if it has been done the results would be worth including or at least commenting on.

Thank you for this interesting comment. Indeed, when membranes are treated with methyl-beta-cyclodextrin to strip cholesterol, caveolae flatten as previously reported in Rothenberg et al., Cell 1992 and Anderson et al., Cell reports 2021 (fig. 4). Below, we show a representative example TEM image of MEF plasma membrane sheets after methyl-beta-cyclodextrin treatment (labeled MbCD). When we analyzed the percentage of low, medium, or highly curved caveolae, we observed for three cell types an increase in both low and highly curved caveolae. However, in the current manuscript, we did not proceed with this drug treatment as cholesterol removal results in other cellular changes including large changes in the clathrin system (Sochacki et al. Dev Cell 2021). In our work we preferred to use the more common and physiological approach of osmotic swelling. We agree that for future experiments this would be an interesting approach to study caveolae which we now mention in the discussion (line 355-357). Likewise, tissue-based changes in the cholesterol system, diseases that modulate cholesterol, and drugs that lower cholesterol would be interesting future directions for study.

4. Figure S6 shows blots of the pacsin and EHBP1 proteins in different knockdown and knockout cells. However EHD2 is not included in any of these blots. I realise the EHD2 KO line was generated in a previous study, but it would be good to show in these particular cells that EHD2 is no longer expressed. Also there does not appear to be a Fig. S6C showing CAV1 and cavin1 levels in the version I have reviewed.

We now include a western blot showing that EHD2 is removed in EHD2 KO cells and have added the caveolin1 and cavin1 western blot which was missing in the previous version of the manuscript (new suppl. fig. S11). We apologize for this mistake.

5. This may merely be my own biased view of the images, but it often appears that caveolae are associated with filamentous protein densities. Are these microtubules? And is there any sense that caveolae (perhaps in contrast to clathrin-coated vesicles) are mostly found in association with these structures?

Cytoskeletal filaments including microtubules and actin are regularly associated with caveolae. In Pt replicas actin filaments (ca. 8 nm) are thinner than microtubules (ca. 25 nm) and are more often observed close to caveolae. The role of actin in caveolae mechano-sensing and adaptation were reported previously, the reviews by Echarri & del Pozo (JCB 2015) and del Pozo et al. (Current Opinion in Cell Bio 2021) nicely summarized current work on caveolae and actin. Detailed caveolae-microtubules contacts were reported by Richter et al. (Traffic 2008). In our work, we did not follow actin or microtubules as the central focus was on coat and coat-associated factors. We agree with the reviewer that in future studies actin and microtubules should be studied in Pt replicas.

6. In Figure 1, when quantifying the number of stripes and the length of stripes, it would be good to see a zoomed example (perhaps as supplementary) of one of the measured stripes to provide a visual representation as to how the cut-offs between the beginning and end of the stripes are determined. I realise this will be somewhat subjective, but it would be useful to see an example. Also would it be possible to provide a measurement of either the stripe widths, or at least the spacing between adjacent stripe structures? Perhaps by simply drawing a line through a few representative caveolae and plotting the intensity versus distance and calculating an average distance between peaks?

Thank you for this comment. We now included a new supplemental figure (fig S2) showing how the length of the coat stripes were measured. Additionally, we now include an estimation of the stripe width as suggested (suppl. fig. S2B, C). As the Pt coating adds approx. 5-6 nm of thickness per stripe we measured the spacing of neighboring stripes (see examples in suppl. fig. S2B). On average, the mean distance between them was 16.2 ± 0.5 nm.

Minor points:

7. Pg. 4, line 133. 'Morphologically' should be 'morphological'.

Thank you, we changed this.

8. In Fig. 2A what is the difference between a 'flat assembly' and 'flat' caveolae?

Based on the comments we have changed this terminology. Specifically, we removed the "flat assembly" state from our figures. These unstructured caveolae vary in size and shape making it difficult to extract quantitative data. Furthermore, without temporal data we cannot identify them as assembling or disassembling. Instead, we focus on clearly identifiable caveolae. Second, we changed the terminology for structured caveolae to "low, medium and highly" curved throughout

the manuscript. We hope this simplifies and clarifies the classification and terminology of the different caveolae types.

9. Pg. 6, line 214. 'in and increase' should be 'in an increase'.

Changed.

10. Pg. 8, line 305. 'with an mild' should be 'with a mild'.

Thank you, we changed this.

Reviewer #3 (Remarks to the Author):

This manuscript aims to describe the formation of caveolae and assign a molecular signature to each of the 3 stages identified. They mainly use a STED-PREM correlative workflow to gather their data.

The authors have acquired an impressive amount of data using this advanced CLEM method to be able to generate quantitative data.

We thank the reviewer for their careful reading and evaluation of our manuscript. Below, please find a detailed responses to all points.

Comments:

- The very first section of the Introduction is very much based on reviews rather than original publications. e.g. the generation of cave-in-1 KO mice.

Due to the large amount of original studies and length constraints, we are not able to include all original articles as references. However, we tried to include as many related newly published studies and reviews as possible to provide a wide context for the reader. We apologize for that we could not include more. However, we have now included additional references to primary papers.

- Page 3, line 80 refers to EM studies but no references are mentioned.

Thank you for this comment. Some representative examples have now been included.

- Page 4, line147. I was surprised to see that the spheres are actually not grape-shaped as in traditional TEM images. Is this a consistent theme or just in this example. It may be good to indicate arrows the overview tomogram to highlight the zoomed areas.

During the preparation of replicas the unroofed membranes are coated with platinum and carbon at an angle. This leads to a slight accumulation of Pt at the base of structures which may lead to slightly different shapes of highly curved caveolae in comparison to thin section TEM of stained membranes. Similar coatings occur for docked dense-core vesicles of PC12 cells (Prasai et al. Nature Comm 2021).

- Page 4, line 154 sounds a bit like this is a new finding in this paper which is well known. Might want to rephrase.

Thank you very much for this comment, we changed the sentence and added references (now line 170-171).

- Page 5, Line 182: associated

Thank you, we changed this accordingly.

- Page 5, Line 187: What about the unlabelled "caveolar-like structures" in first zoom just above and below Cathrin labelled structures. The fluorescence for cavin in zoom "a" actually looks shifted.

Membranes can contain organelles that have a shape and size similar to caveolae but lack the coat. Without CLEM these are challenging to identify. They could be other endocytic or exocytic organelles. Shifting of the fluorescence signal compared to the underlying membrane can occur during preparations steps between light and electron microscopy. In particular, highly curved caveolae can contain a small and flexible neck which may allow the organelle to shift slightly during EM preparation in relation to the fluorescence. Yet, the majority of correlated caveolae fluorescence match the underlying membrane shapes in EM. We have shown in past work for various organelles that these errors are small (Sochacki et al. 2014, 2017, Prasai et al. 2021). We now more clearly discuss these potential errors and issues in the text (line 330-334).

- Page 5, Line 188, Fig 4A,B: These images contradict the statement that the size of caveolae in STED is the same as PREM. STED fluorescence is clearly wider distributed. Not just at the formation stage.

Thank you for this comment. We agree that STED signals sometimes do not match exactly to their underlying membrane domains at the single vesicle level at this resolution. This could be caused by the size differences due to primary and secondary antibodies or slight shifts in the images. The measured size is based on a half width of the fluorescence data. However, when the average mean values of STED and EM are compared, we do not detect gross differences indicating that STED depletion of the Atto647N dye resolves caveolae size surprisingly well (see suppl. fig. 4d).

- Page 6, Line 209: First images of figures 4, 5, 6B do not show correct scale bar. Would also have been "nice" to have a consistent way of depicting those.

We apologize for this mistake and corrected and updated the scale bars. We also tried to unify scale bars throughout the manuscript.

- Page 7, Line 231: Although this is briefly discussed later: As caveolae are dynamic how can a distinction be made between forming and collapsing / merging structures. Is the protein composition the same for both. As mentioned by the authors, mild osmotic shock and addition of

lipids could discriminate between the 2. This is something that is not really addressed by the authors.

We thank the reviewer for this important comment. Indeed, we cannot distinguish between newly formed or collapsed/dissembled caveolae. Therefore we now excluded the “assembly or immature” state from the main text and analysis. To discriminate between those two states temporal information would be needed, which is not currently possible with our methods. Future experiments with live cells (or with a factor that can distinguish these states) are needed to dismantle this process.

- Page 7, Line 260: I disagree with the conclusion of the authors that there is no Dynamic colocalization with caveolae. I see a clear colocalization of the Dynamic K44A mutant in the Zoom sections. The argument that those would be Cathrin structures because of the few yellow spots would a very selective interpretation.

This last point is my main issue with this manuscript.

Otherwise the manuscript is well written, has a logical flow with very impressive quantitative imaging data.

Our STED-only analysis indicates minimal colocalization of the mutant dynaminK44a and caveolae (~18% of all inspected caveolin1 spots are dyn-k44A positive). Even less wild-type dynamin was detected at caveolae with STED. The same observation was recently described by Larsson et al., (Biorxiv 2022, Fig. 5). To confirm these findings, we now used dSTORM-CLEM to precisely localize dynamin in EM images because dSTORM resolution is higher and more punctate than STED (suppl. fig. 13). In these experiments, caveolae in EM were very rarely (below 8% of all inspected caveolae) positive for dynamin. At the plasma membrane, caveolae are often close to clathrin (example CLEM images below). We propose that this pairing may lead to the higher number of colocalization events between caveolae and dynamin in the fluorescence images. For instance in HeLa, STORM-CLEM analysis revealed that 75.3% of dynamin stained caveolae are close to clathrin (less than 50 nm distance, suppl. fig. S13). Several CLEM examples are shown below where clathrin (honeycomb-like) and caveolae (watermelon-like) are closely positioned and the dynamin signal (magenta) is between the clathrin and caveolae. These pairings could result in mis-assignment of a protein to one or the other structure in fluorescence images. To further investigate dynamin localization to caveolae, however, we now include STED microscopy of dynamin antibody staining after caveolae endocytosis was triggered by fatty acid treatment (new suppl. fig. S12d). This dynamin antibody strongly localizes to clathrin (fig. S12c CLEM, fig S12e STED). When colocalization to caveolae was tested, we again did not detect strong dynamin localization. In summary, we suggest based on our data, that dynamin very rarely localizes to caveolae. At the reviewers suggestion, however, we now state that dynamin could still interact with caveolae but the temporal association would need to be very fast or very sparsely localized. We hope this interpretation is more balanced.

HeLa: Dynamin2 antibody
3 cells, 852.6 μm^2
11 clathrin caveolae close distance
($< 50 \text{ nm}$ distance)

HeLa: Dynamin2-GFP
3 cells, 865.1 μm^2
19 clathrin caveolae close distance
($< 50 \text{ nm}$ distance)

REVIEWERS' COMMENTS

Reviewer #1 (Remarks to the Author):

This revised version addresses all our concerns and provides sufficient evidences to sustain the authors' claims. This is a beautiful study providing important information about caveolae biology/plasticity. However, there are some minor issues that should be addressed.

1. In line 127-128 the authors refer (I believe) to figure 1d,e with the following statement. "They also associate with small and disorganized coat domains with low curvatures". It would be helpful to mark in the images to which specific caveolae they refer to. It is hard to see which of the shown coats are disorganized and which ones are organized.

2- Line 200. The authors state "To test this hypothesis,"; to which one do they refer to? To the hypothesis of the previous line/end of previous section (line 197-8)? Please, organize the transition in these two sections.

3- Line 220-21: "As shown previously⁴⁰, osmotic shock in HUVEC resulted in an increase in lower curved caveolae independent of the intensity of osmotic shock". This sentence needs to be rewritten. Sinha et al. (ref. 40) observed less caveolae (by EM and TIRF) and presumably (did not quantified) more flatten caveolae by Deep-etched EM with osmotic shock; however, they did not measure lower curved caveolae. I understand that it is difficult to draw a line between less curved and flatten, but the field is already confused, so using similar terminology may help us moving forward.

Matthaeus et al. Response to Reviewer

We thank the reviewers for their helpful comments and suggestions. We addressed all concerns and revised our manuscript accordingly. Please see below our response.

REVIEWERS' COMMENTS

Reviewer #1 (Remarks to the Author):

This revised version addresses all our concerns and provides sufficient evidences to sustain the authors' claims. This is a beautiful study providing important information about caveolae biology/plasticity. However, there are some minor issues that should be addressed.

We thank the reviewer for their positive review of our manuscript. We revised our manuscript based on the comments below.

1. In line 127-128 the authors refer (I believe) to figure 1d,e with the following statement. "They also associate with small and disorganized coat domains with low curvatures". It would be helpful to mark in the images to which specific caveolae they refer to. It is hard to see which of the shown coats are disorganized and which ones are organized.

Thank you. We added precise description (line 121).

2- Line 200. The authors state "To test this hypothesis,"; to which one do they refer to? To the hypothesis of the previous line/end of previous section (line 197-8)? Please, organize the transition in these two sections.

We changed the sentence accordingly (line 193).

3- Line 220-21: "As shown previously⁴⁰, osmotic shock in HUVEC resulted in an increase in lower curved caveolae independent of the intensity of osmotic shock". This sentence needs to be rewritten. Sinha et al. (ref. 40) observed less caveolae (by EM and TIRF) and presumably (did not quantified) more flatten caveolae by Deep-etched EM with osmotic shock; however, they did not measure lower curved caveolae. I understand that it is difficult to draw a line between less curved and flatten, but the field is already confused, so using similar terminology may help us moving forward.

We agree with the reviewer and changed the sentence to be more precise here.

Reviewer #2 (Remarks to the Author):

The reviewers have answered all of my queries. I have read the other reviewer comments and responses, and feel that the authors have sufficiently addressed any substantive criticisms.

We thank the reviewer for their positive review.